# EMBO *reports*

# Regulated microexon alternative splicing in single neurons tunes synaptic function

Bikash Choudhary[1], Rebekah Napier-Jameson [ID] [2] & Adam Norris [ID] [1][✉]

## Abstract

Microexons are important components of the neuronal transcriptome. Though tiny, their splicing is essential for neuronal development and function. Microexons are typically included in the nervous system and skipped in other tissues, but less is known about whether they are alternatively spliced across neuron types, and if so what the regulatory mechanisms and functional consequences might be. We set out to globally address this question in *C. elegans* using deep single-cell transcriptomes and in vivo splicing reporters. We find widespread alternative microexon splicing across neuron types. Focusing on a broadly-conserved 9-nucleotide exon in the synaptic vesicle gene *unc-13*, we find that it is completely skipped in olfactory neurons, but completely included in motor neurons. This splicing pattern is established by two neuronal RNA binding proteins which recruit spliceosomal component PRP-40 to mediate microexon inclusion. Cell-specific microexon alternative splicing is functionally important, as forcing microexon inclusion causes olfactory defects, while forcing microexon skipping causes locomotory defects. These locomotory defects are caused by decreased inhibitory motor neuron synaptic transmission and altered synaptic vesicle distribution. Regulatory features of *unc-13* microexon splicing are broadly conserved: related MUN-domain genes in worms, flies, and mice invariably encode microexons, and those we tested are subject to similar regulatory principles (*e.g.* included in motor neurons, skipped in olfactory neurons, and regulated by the same two RNA binding proteins). Thus, not only is microexon inclusion important for nervous system function, but microexon alternative splicing across neurons is important for tuning neuronal function in individual cell types.

**Keywords** Splicing; Microexon; *unc-13*; Alternative Splicing; Neuron
**Subject Categories** Neuroscience; RNA Biology

## Introduction

Alternative splicing is the process by which sequences in pre-mRNA can be selectively included or excluded from the mature mRNA. This process is an important source of gene regulation as well as transcriptomic and proteomic diversity (Carvill and Mefford, 2020; Chen and Manley, 2009; Lin et al, 2020; Marasco and Kornblihtt, 2023). The molecular diversity generated by alternative splicing is important for establishing functional heterogeneity across cell types, most notably among neurons (Furlanis et al, 2019; Marasco and Kornblihtt, 2023; Norris and Calarco, 2012). Recent studies have begun to identify regulatory and functional aspects of alternative splicing at the resolution of individual neuron types (Ciampi et al, 2022; Feng et al, 2021; Thompson et al, 2019; Zhang et al, 2016).

One striking feature of neuronal transcripts is the abundance of very small exons referred to as microexons, often defined as exons ≤30 nucleotides (nt) in length (Volfovsky et al, 2003). Microexons are enriched in neuronal transcripts, tend to be included in neurons but skipped in other tissues, and are dysregulated in neurological disorders such as autism spectrum disorders (Gandal et al, 2018; Irimia et al, 2014). Recently, functional roles for individual microexons have been identified, including microexons required for neuronal development, behavior, learning, and memory (Gonatopoulos-Pournatzis et al, 2020; Johnson et al, 2019; Lin et al, 2020; Nakano et al, 2018; Quesnel-Vallières et al, 2015).

Microexon inclusion is thus essential for proper neuronal development and function, but do microexons also undergo alternative splicing within the nervous system? The observation of microexon inclusion in neurons and skipping in other cell types may yet accommodate additional heterogeneity of microexons that are included in certain neuron types and skipped in other neuron types. Technical limitations have made this question difficult to address due to the small size of microexons and the limited depth and capture efficiency inherent to technologies such as single-cell sequencing (Parada et al, 2021). Nevertheless, recent global transcriptomic analyses with cell-specific and brain-region-specific resolution provide evidence for the existence of alternative microexon splicing across neuronal cell types (Ciampi et al, 2022; Han et al, 2024; Koterniak et al, 2020; Parada et al, 2021). Therefore, the regulatory and functional properties of such neuron-type-specific microexon alternative splicing is of great interest.

We set out to systematically identify microexon alternative splicing across individual neurons, to discover regulators of neuron-type-specific microexon splicing, and to determine functional roles for individual microexons in individual neuron types. We leveraged the strengths of *C. elegans* to perform in vivo imaging of microexon splicing with single-neuron resolution, complemented by deep single-neuron RNA-Seq data generated by the CeNGEN consortium (Barrett et al, 2025; Wolfe et al, 2025).

[1]Department of Biochemistry, University of California, Riverside, 3401 Watkins Drive, Boyce Hall, Riverside, CA 92521, USA. [2]Department of Biological Sciences, Southern Methodist University, Dallas, TX 75205, USA. [✉]E-mail: adamn@ucr.edu

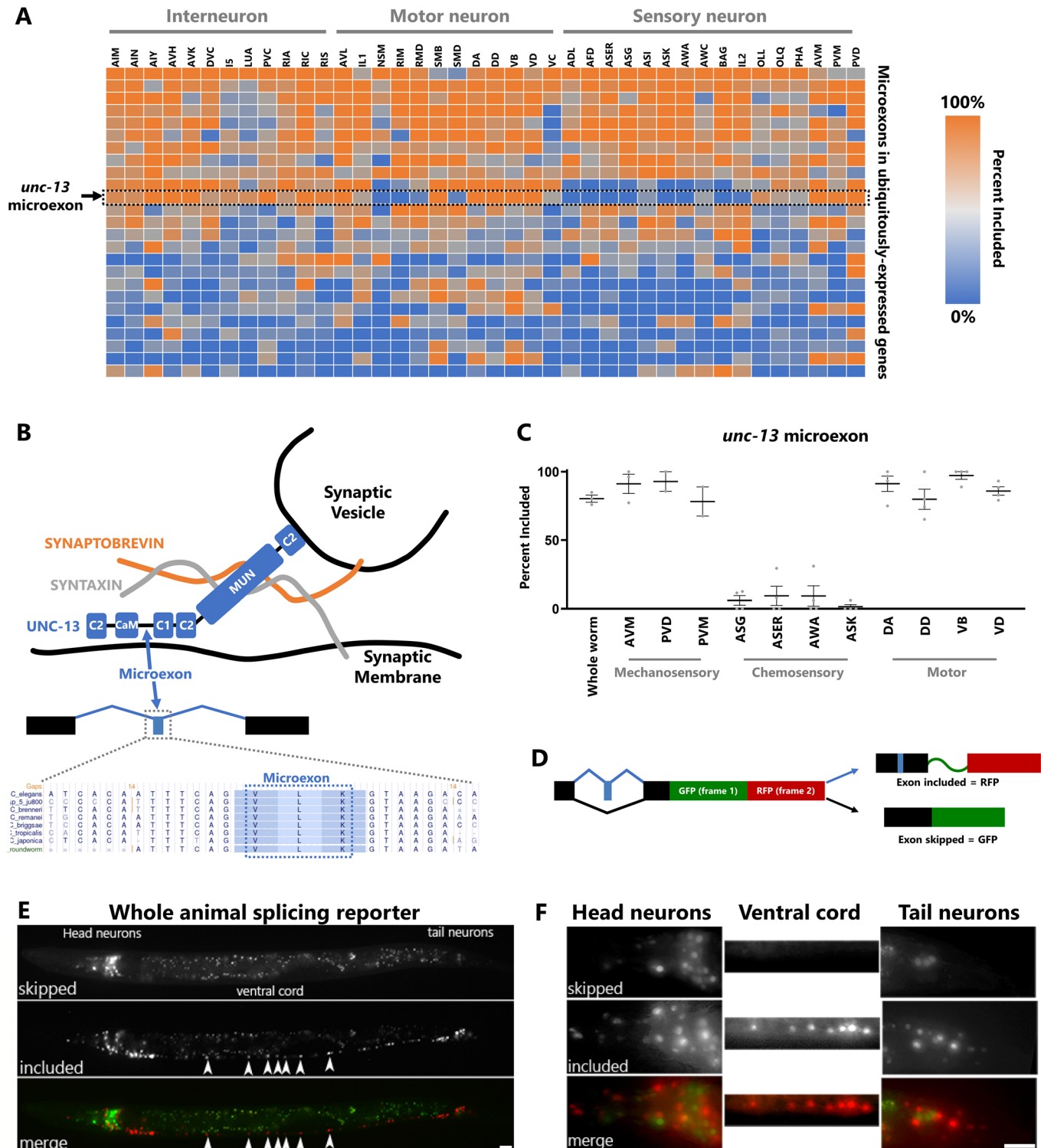

**A** Interneuron / Motor neuron / Sensory neuron

*unc-13* microexon

Microexons in ubiquitously-expressed genes

Percent Included 100% / 0%

**B**

Synaptic Vesicle

SYNAPTOBREVIN
SYNTAXIN

UNC-13 C2 CaM C1 C2 MUN C2

Synaptic Membrane

Microexon

**C** *unc-13* microexon

Percent Included

Whole worm | AVM PVD PVM (Mechanosensory) | ASG ASER AWA ASK (Chemosensory) | DA DD VB VD (Motor)

**D** Exon included = RFP / Exon skipped = GFP

GFP (frame 1) RFP (frame 2)

**E** Whole animal splicing reporter

Head neurons / tail neurons

skipped

ventral cord

included

merge

**F** Head neurons / Ventral cord / Tail neurons

skipped

included

merge

Here we focus on a highly conserved 9 nt microexon in the synaptic vesicle gene *unc-13*, which we find is completely included in motor neurons, but completely skipped in olfactory neurons. We identify a pair of RNA-binding proteins that establish this splicing pattern, and investigate the mechanisms by which they recruit the spliceosomal component PRP-40 to mediate microexon inclusion.

Although the microexon encodes a mere three amino acids, forcing the microexon to be skipped causes synaptic transmission defects in motor neurons, while forcing it to be included causes defects in olfactory behavior. Finally, we find that genes in the *unc-13* family (*unc-13/Munc13, unc-31/Cadps*, etc.) invariably contain microexons in both worms and mice, and this family of microexons is subject to

**Figure 1. Microexon alternative splicing across neuronal cell types.**

(A) Heat map displaying % inclusion for all alternatively spliced microexons in pan-neuronally expressed genes. Arrow denotes the *unc-13* microexon studied in this paper. (B) Schematic of UNC-13 protein depicting its conserved domains, relative position of the microexon, and role in the release of synaptic vesicles at synapse. Below is a depiction of microexon conservation across nematode species. (C) Quantification of percent inclusion of *unc-13* microexon in different neuronal subtypes (mechanosensory, chemosensory and motor neurons) from CeNGEN RNA-Seq data (Taylor et al, 2021). n ≥ 2 independent RNA-Seq experiments. Error bars display the standard error to mean (SEM). (D) Schematic of the bicolor splicing reporter. Inclusion of the alternative microexon (in blue) results in RFP expression, while exclusion leads to GFP expression. (E) Representative image of a transgenic animal expressing the *unc-13* microexon bicolor splicing reporter under a pan-neuronal promoter. Ventral cord neurons (white arrowheads) only express the included version of *unc-13* microexon. (F) Splicing pattern of the *unc-13* microexon in different regions of the *C. elegans* nervous system (head neurons, nerve cord, and tail neurons). Note that the wild-type control panel is the same worm as in Figs. 2D and EV4A. Each distinct puncta is an individual cell body. Scale bar 20 μm. Source data are available online for this figure.

shared regulatory principles (e.g., included in motor neurons, skipped in olfactory neurons). These results highlight an important new layer of microexon regulation: not only is microexon inclusion important for nervous system function, but regulated alternative microexon inclusion is important for neuron-specific function.

## Results

### Microexons are alternatively spliced across neuronal cell types

In mammals, microexons are enriched in neuronal transcripts, and tend to be included in neurons but skipped in non-neuronal cells (Irimia et al, 2014). We recently found this to be the case in *C. elegans* as well, and further raised the possibility that not only do microexons tend to be included in neuronal cells, but additional microexon alternative splicing heterogeneity exists between neuronal cell types (Choudhary et al, 2021). To globally examine this question, we analyzed deep-sequenced transcriptomic data obtained from individual neuronal cell types. This data, generated by the CeNGEN consortium, represents 46 of the 118 neuronal cell types in the *C. elegans* nervous system (Taylor et al, 2021).

We found that indeed, inclusion levels of individual microexons are highly variable across neuronal cell types. Many microexons have high inclusion levels throughout the nervous system, but even for these microexons, there are typically a handful of neuronal cell types in which the microexon is mostly skipped. Some microexons are spliced in a binary pattern across cell types (either 100% included or 100% skipped) while others are spliced in a more graded manner across cell types (Fig. 1A).

One particularly striking pattern of microexon alternative splicing is in the gene *unc-13*, which encodes a critical regulator of synaptic vesicle exocytosis (Richmond et al, 1999). The microexon is 9 nt long, and is unusually highly conserved. The amino acid sequence is invariant across nematode and *Drosophila* species, and a similar 6 nt microexon resides in the mammalian *Unc13c* gene at the identical position, located between the regulatory calmodulin binding (CaM) and C1 domains (Figs. 1B and EV1A).

In *C. elegans*, the microexon is spliced in a binary manner that is strongly correlated with cell type. For example, in various types of mechanosensory neurons and motor neurons the microexon is nearly 100% included, while in various types of chemosensory neurons, it is nearly 100% skipped (Fig. 1C). Thus, although the *unc-13* microexon is primarily included across the nervous system (~75% included), it is under tight regulatory control such that certain cell types express only the microexon-skipped isoform.

To further validate these findings in vivo, we generated a fluorescent two-color splicing reporter for the *unc-13* microexon. The reporter is constructed so that if a cell produces the included isoform, RFP will be translated in-frame, whereas if a cell produces the skipped isoform, GFP will be translated instead (Fig. 1D) (Norris et al, 2014). We expressed the reporter under a pan-neuronal promoter and found that most neurons express the included isoform (RFP), whereas a small subset of neurons express the skipped isoform (GFP) (Figs. 1E,F and EV1B). In agreement with the transcriptomic data, we found that individual cell types make binary splicing decisions: most cells express only GFP or RFP, with very few cells expressing both (Fig. EV1C).

The splicing reporter also confirms the cell-specific splicing transcriptomic observations. For example, the motor neurons of the ventral nerve cord all exhibit 100% microexon inclusion, as do mechanosensory neurons, while the chemosensory neurons in the head exhibit 100% microexon skipping (Fig. 1C,F). Together, these results indicate that microexons are not only included in the nervous system compared to other tissues, but also subject to strict regulation across cell types within the nervous system. Given the striking cell-type specificity of the *unc-13* microexon, we have used it as a model for understanding the regulation and function of such neuron-specific microexon alternative splicing.

### Cell-specific splicing of the *unc-13* microexon is established by neuronal RBPs EXC-7 and MBL-1

We next asked how the striking cell-specific alternative splicing pattern of *unc-13* might be generated. We previously showed that PRP-40, a component of the U1 spliceosome, is essential for the inclusion of nearly all microexons in *C. elegans*, and indeed it is required for *unc-13* microexon inclusion (Fig. 2A). However, PRP-40 alone is not sufficient to explain the striking alternative splicing patterns, as it is expressed ubiquitously and is required for microexon inclusion in all cells (Choudhary et al, 2021).

To search for regulatory *trans*-acting factors, we began by analyzing existing RNA-seq data from animals with loss-of-function mutations in RNA-binding protein (RBP) genes expressed in the *C. elegans* nervous system. While most mutants do not affect *unc-13* microexon inclusion, we discovered that *exc-7/Elav/Hu* and *mbl-1/Mbnl1* single mutants result in small decreases in *unc-13* microexon inclusion. In contrast to the modest changes observed in single mutants, RNA Seq from *exc-7; mbl-1* double mutants reveals that *unc-13* microexon inclusion is completely abolished, just as in *prp-40* mutants (Fig. 2A,B). RT-PCR analysis agrees with the RNA Seq data, showing that microexon inclusion decreases in *mbl-1* or *exc-7* mutants, and that *exc-7; mbl-1* double mutants have a

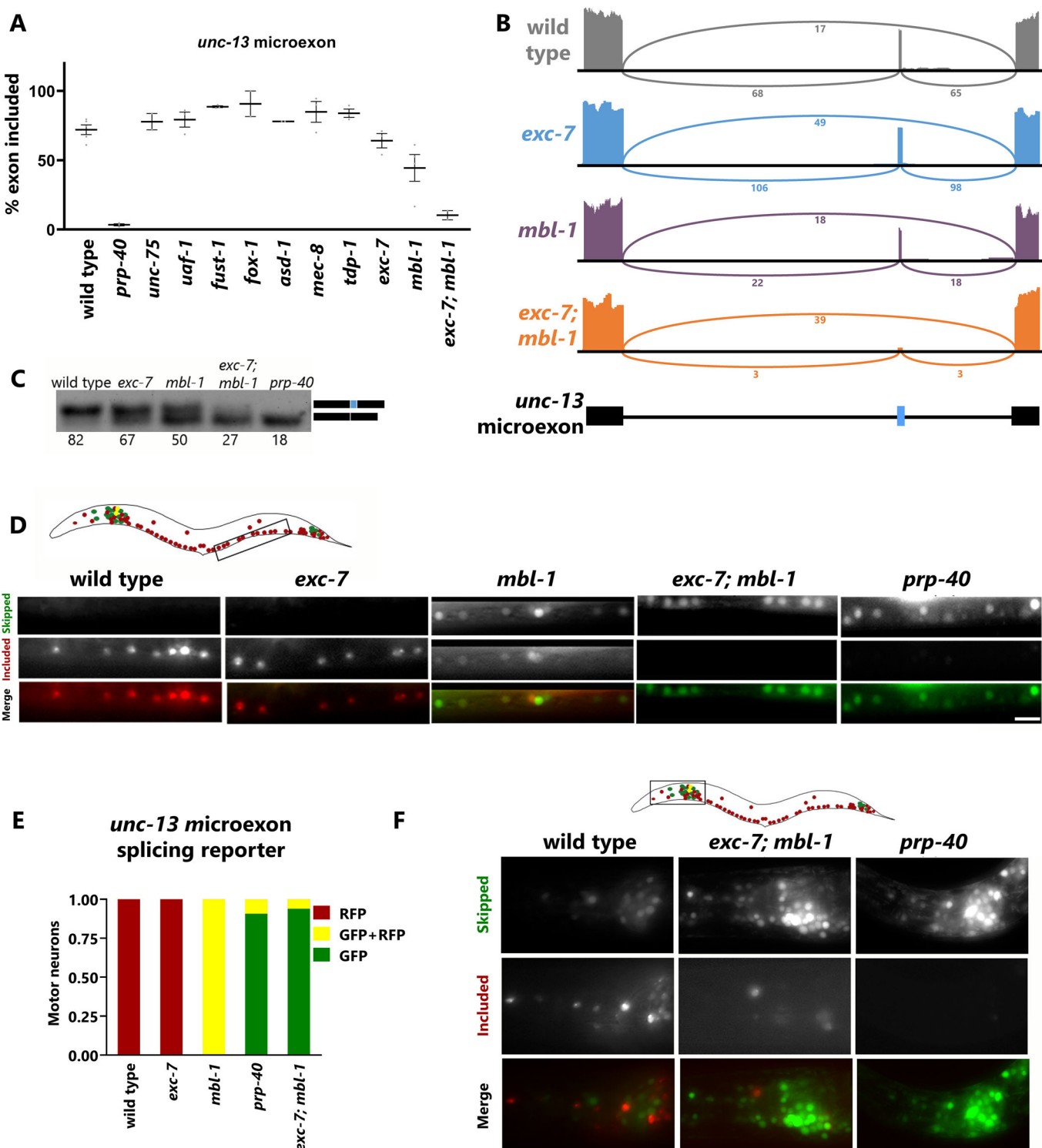

complete loss of microexon inclusion, similar to that of *prp-40* mutants (Fig. 2C).

To assess how EXC-7 and MBL-1 affect microexon inclusion with neuronal-subtype specificity, we crossed the *unc-13* microexon splicing reporter into single and double-mutant backgrounds. We focused on motor neurons of the ventral nerve cord, which exclusively express the included isoform (RFP) and are easy to score and identify due to their linear arrangement along the nerve cord (Fig. 2D) (Norris et al, 2014). We found that the *mbl-1* mutation results in a decrease in microexon inclusion for most neurons. Many neurons that exhibit 100% inclusion in the wild type shift to a partial inclusion phenotype in *mbl-1* mutants

◄ **Figure 2. Cell-specific splicing of *unc-13* microexon is regulated by the neuronal RBPs EXC-7 and MBL-1.**

(A) Percent inclusion for *unc-13* microexon in existing RNA Seq data for mutants of neuronal RBP genes. $n \geq 2$ independent RNA Seq experiments. All mutants are predicted to be molecular nulls (deletions) except for *uaf-1*, which is a hypomorphic allele. Error bars display the SEM. (B) Sashimi plots of *unc-13* microexon from whole animal RNA Seq data for wild type, *exc-7 (csb28)*, *mbl-1 (csb31)* and *exc-7 (csb28); mbl-1 (csb31)*. (C) RT-PCR of *unc-13* microexon alternative splicing in mutant genotypes. Upper band is microexon included, lower band is microexon skipped. Numbers below each band represent % inclusion values determined by gel densitometry. (D) *unc-13* microexon splicing reporter in ventral cord neurons (neuron locations demarcated in cartoon above) in wild type, *mbl-1 (wy560)*, *exc-7 (rh252)*, and *prp-40 (csb3)* mutants. Note that the wild-type control panel is the same worm as in Figs. 1F and EV4A. (E) Quantification of the splicing pattern in ventral cord neurons in the genotypes indicated, $n = 10$–$15$ animals. (F) Representative images of head neurons for wild type, *exc-7 (rh252); mbl-1 (wy560)* and *prp-40 (csb3)* animals expressing *unc-13* microexon-splicing reporter. Inclusion of microexon is almost completely lost in both *exc-7 (rh252); mbl-1 (wy560)* and *prp-40 (csb3)* genotypes. Scale Bar 10 µm. Source data are available online for this figure.

(Fig. 2D,E). In contrast, *exc-7* mutants result in no detectable changes in microexon splicing in motor neurons, though a handful of head neurons exhibit decreased inclusion (Fig. EV2A). However, *exc-7; mbl-1* double mutants result in complete microexon skipping in most neurons, including all neurons of the ventral nerve cord (Fig. 2D–F). The splicing phenotype in motor neurons is more extreme in double mutants than would be predicted by simply combining the single mutant phenotypes (no phenotype in *exc-7* and modest phenotype in *mbl-1*), suggesting that EXC-7 and MBL-1 cooperate to facilitate *unc-13* microexon inclusion.

*exc-7; mbl-1* double mutants result in a complete loss of microexon inclusion across cell types, indicating that the combined activity of these two factors establishes the cell-specific microexon splicing patterns seen in wild-type animals. The splicing patterns in these double mutants are similar to *prp-40* mutants (Fig. 2D–F), which exhibit an overall loss of inclusion across the nervous system. This suggests the possibility that EXC-7 and MBL-1 confer cell-specific microexon splicing by facilitating the activity of PRP-40 in specific cell types.

## EXC-7 and MBL-1 *cis* elements downstream of *unc-13* microexon mediate cell-specific splicing

To further probe the regulatory mechanisms by which EXC-7 and MBL-1 mediate microexon inclusion, we searched for *cis*-regulatory motifs present in the *unc-13* pre-mRNA that confer cell-specific regulation. Because the small size of microexons likely precludes the presence of exonic regulatory elements (Ustianenko et al, 2017), we tested whether cell-specific regulatory elements are present in the intronic regions surrounding the *unc-13* microexon. We first tested the effects of deleting large portions of either the upstream or downstream introns. We found that the upstream deletion has no effect on *unc-13* splicing, but the downstream deletion causes an increase in microexon skipping (Fig. 3A,B).

We next asked whether the downstream intron might harbor binding elements for EXC-7 and MBL-1. We searched for evolutionarily conserved sequences matching the in vitro-derived consensus motifs determined for the two factors (Ray et al, 2013), and identified two putative elements for both EXC-7 (UUAAGUU) and MBL-1 (GCUUGC) (Fig. 3C). We tested these *cis* motifs for regulatory activity by generating splicing reporters mutated for either the EXC-7 *cis* motifs, the MBL-1 motifs, or both motifs combined (Fig. 3D,E). Mutating the MBL-1 sites causes decreased microexon inclusion, similar to *mbl-1* mutants. Mutating the EXC-7 sites does not affect splicing in motor neurons, but does cause decreased inclusion in a small number of head neurons, similar to *exc-7* mutants (Fig. EV3A).

Simultaneously mutating both the EXC-7 and MBL-1 sites results in a stronger splicing phenotype than either the MBL-1 or EXC-7 single

mutations in isolation, though not as severe as in the *exc-7; mbl-1* double mutant (Figs. 3D,E and EV3B). Taken together, these results suggest that EXC-7 and MBL-1 combinatorially regulate the *unc-13* microexon via adjacent binding sites in the downstream intronic region.

## *prp-40* and *mbl-1* interact to regulate cell-specific microexon inclusion

We hypothesized that upon binding to the downstream intron of *unc-13*, MBL-1 and EXC-7 recruit PRP-40 to mediate microexon inclusion, thus providing a mechanistic basis for neuron-type-specific microexon inclusion. To test whether PRP-40 physically interacts with MBL-1, we performed co-immunoprecipitations using strains with endogenous CRISPR-mediated epitope tags. Immunoprecipitation of MBL-1::3x-FLAG using anti-FLAG antibodies robustly and specifically co-immunoprecipitates PRP-40::mScarlet (Fig. 3F, right lane). No such co-immunoprecipitation is observed in the control condition where MBL-1 is untagged (Fig. 3F, left lane). We previously performed similar co-immunoprecipitation experiments and found that EXC-7 and PRP-40 likewise co-immunoprecipitate (Choudhary et al, 2021). These results are consistent with the model of MBL-1 and EXC-7 binding to the downstream intron and recruiting PRP-40.

We further tested this model genetically using double-mutant analysis. We find that the *prp-40* mutant phenotype (complete microexon skipping) is dominant over either of the *exc-7* or *mbl-1* mutant phenotypes (Figs. 3G and EV4A), consistent with our model that PRP-40 recruitment is the necessary downstream molecular event required for microexon inclusion upon RBP binding. Next, we found that overexpression of PRP-40 does not rescue *exc-7; mbl-1* double-mutant phenotypes (Figs. 3H and EV4A), indicating that increased PRP-40 levels cannot bypass the requirement for EXC-7 and MBL-1 activity. This is again consistent with the model that EXC-7 and MBL-1 are required to direct the activity of PRP-40 to the *unc-13* microexon through their sequence-specific binding to the downstream intron.

We also tested microexon splicing phenotypes in trans-heterozygote conditions. If a phenotype is observed in a strain heterozygous for recessive mutations in two different genes, this often indicates that the two genes are components of the same molecular pathway and/or the same physical complex (Choudhary et al, 2021; Yook et al, 2001). Double heterozygotes for *mbl-1* and *exc-7* are indistinguishable from wild type (Figs. 3H and EV4A), but double heterozygotes for *mbl-1* and *prp-40* exhibit decreased microexon inclusion (Figs. 3H and EV4A–C), suggesting a shared molecular pathway between MBL-1 and PRP-40. These defects are similar in nature to the MBL-1 *cis* element mutations (Fig. 3E). Taken together, our genetic and biochemical experiments support

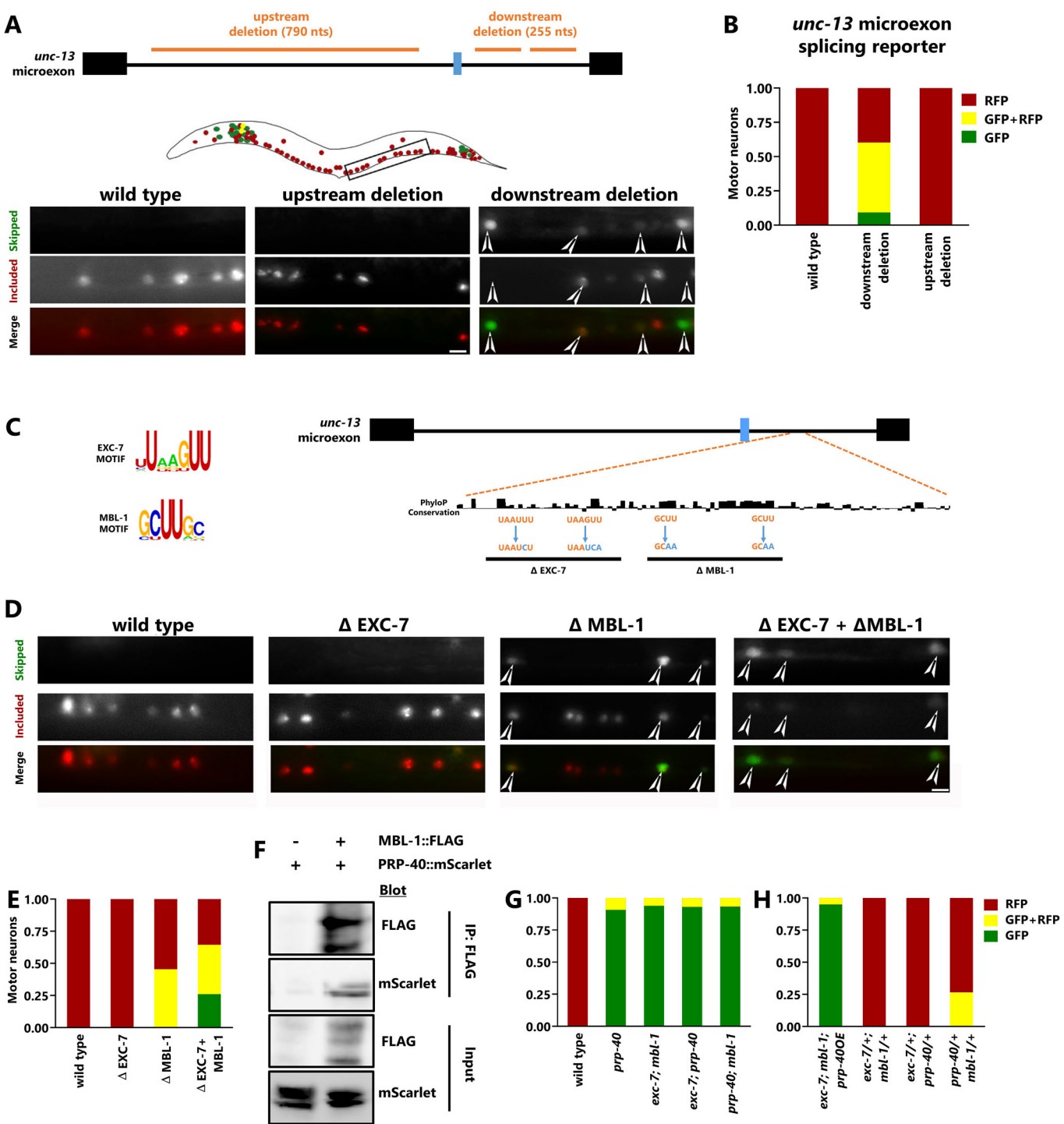

the notion that EXC-7 and MBL-1 recruit PRP-40 to the *unc-13* microexon to facilitate neuron-specific splicing.

## *unc-13* microexon skipping is required for proper olfactory behavior

Having established that the *unc-13* microexon is strictly regulated in a cell-type-specific manner, we next asked whether this regulation has functional consequences in individual neurons. UNC-13 is a highly conserved presynaptic protein that interacts with the neuronal SNARE complex to facilitate synaptic vesicle release (Dittman, 2019). Loss of *unc-13* in *C. elegans* prevents synaptic transmission and results in severe defects in locomotion (Kohn et al, 2000). The *unc-13* microexon is situated between the CaM and the C1 domain (Fig. 4A), both of which carry out modulatory roles affecting the activity of the core MUN domain

◀    **Figure 3.    Regulatory mechanisms underlying cell-specific alternative splicing of *unc-13* microexon.**

(A) Upper panel: Schematic of *unc-13* microexon with surrounding introns and constitutive exons, orange lines denote transgene deletions. Conserved regions we hypothesized to be necessary for basal splicing machinery were left undeleted, as displayed (for this reason, the downstream intron is a bipartite deletion of 255 nts). Lower panel: representative images of motor neurons expressing *unc-13* microexon splicing reporter with intronic deletions. White arrowheads indicate the appearance of GFP in ventral cord neurons. Downstream intron deletions, but not upstream deletions, cause increased microexon skipping. (B) Quantification of the splicing pattern in ventral cord neurons expressing the modified *unc-13* microexon splicing reporter of Fig. 3A. (C) Left panel: in vitro-derived motifs for EXC-7 and MBL-1 as reported from RNA compete data. Right panel: EXC-7 and MBL-1 *cis* elements present in the downstream intron. PhyloP conservation scores are displayed, and the *cis* element mutations generated are marked in blue. (D) Representative images of ventral cord neurons expressing the abovementioned *cis*-element-mutated splicing reporters. The white arrowhead marks the appearance of GFP in the ventral cord neurons. (E) Quantification of experiments displayed in 3D. (F) Biochemical interaction of PRP-40::mScarlet with MBL-1::Flag. MBL-1::Flag was pulled down by anti-flag, and the blot was probed for PRP-40::mScarlet (right lane). As a control, PRP-40::mScarlet lysate was incubated with anti-flag (left lane). (G, H) Quantification of genetic interaction (G) and double heterozygote analysis (H) by quantifying splicing reporter in the ventral cord, as in panel (E). Also refer to Fig. EV4. Graphs, $n = 10$–$15$ animals. Scale Bar 10 μm. Source data are available online for this figure.

(Lipstein et al, 2012; Michelassi et al, 2017). We therefore hypothesized that the presence or absence of a microexon in this location might modulate UNC-13 activity.

To explore the functional role of the *unc-13* microexon we made endogenous edits to the genome that either forced microexon inclusion ($UNC$-$13_{included}$) or microexon skipping ($UNC$-$13_{skipped}$) (Fig. 4B). These edits do not abrogate the function of UNC-13, as phenotypes such as pharyngeal pumping are unaffected, in contrast with strong reductions in a loss-of-function *unc-13* mutant (Fig. 4C). Moreover, addition of an endogenous C-terminal GFP tag to the isoform-specific edited strains shows that the edits do not affect expression levels or presynaptic localization of the UNC-13 protein (Fig. 4D,E). Similarly, when isoform-specific *unc-13* cDNAs are over-expressed in the context of a null *unc-13* mutant, both the microexon-included and microexon-skipped transgenes rescue the strong locomotory phenotype (Fig. 4F). Together, these results show that both isoforms of UNC-13 represent functional products.

To ask whether specific neuron types require *unc-13* microexon skipping, we first focused on neurons that exclusively express the microexon-skipped isoform of *unc-13*. Cell-specific RNA Seq indicates that chemosensory neurons uniformly express the *unc-13* microexon-skipped isoform, and our splicing reporters agree with this (Figs. 1C and 4G). To further confirm this observation in a single neuron, we generated an *unc-13* splicing reporter expressed only in the AWA chemosensory neuron under the *odr-10* promoter, which yielded 100% GFP (exon skipping) in the AWA neuron (Fig. 4G).

We therefore tested whether the AWA function requires the UNC-13 skipped isoform for proper function. The AWA olfactory neuron is necessary for chemotaxis toward attractive volatile chemicals, including pyrazine (Liang et al, 2022). We performed pyrazine chemotaxis assays and found that while the $UNC$-$13_{skipped}$ strain chemotaxes appropriately, the $UNC$-$13_{included}$ strain is nearly achemotactic, with animals distributed evenly throughout the plate irrespective of the location of the olfactant. The phenotype is nearly as strong as the achemotactic positive control mutant *che-2* (Fig. 4H). We therefore conclude that appropriate olfactory behavior controlled by the AWA neuron requires *unc-13* microexon skipping.

## *unc-13* microexon inclusion is required for locomotory behavior and neuromuscular synaptic transmission

We next tested whether other neuron types have a functional requirement for *unc-13* microexon inclusion. We focused on

ventral nerve cord motor neurons, since they express only the microexon-included version of UNC-13 (Fig. 1C,F) and their activity can be easily assayed by behavioral and pharmacological approaches. These motor neurons are responsible for the locomotory behavior of the animal, and are composed of excitatory (cholinergic) and inhibitory (GABAergic) neurons, which coordinate muscle contraction and relaxation resulting in the sinusoidal movements of the animal (Fig. 5A) (Zhen and Samuel, 2015). Simple locomotory assays (swimming/thrashing assays) reveal that $UNC$-$13_{included}$ animals have normal locomotion, as expected, while $UNC$-$13_{skipped}$ animals have modest locomotory defects (Fig. 5B). These defects are much less severe than in *unc-13* null mutants (Fig. 5B), indicating that the skipped isoform of UNC-13 is partially functional in motor neurons, where it is normally not expressed, but it is not sufficient to confer full function to motor neurons lacking the microexon-included isoform.

*unc-13* loss-of-function mutants have strong defects in neurotransmitter release. This can be observed by exposing animals to the drug aldicarb, which causes progressive buildup of the excitatory neurotransmitter acetylcholine at neuromuscular synapses, resulting in muscle hypercontraction and paralysis (Mahoney et al, 2006). Loss of *unc-13* causes resistance to aldicarb (Fig. 5C), as acetylcholine release is deficient, and thus the progressive buildup and subsequent paralysis are delayed. $UNC$-$13_{included}$ animals behave like wild type in the aldicarb assay (Fig. 5C). Surprisingly, $UNC$-$13_{skipped}$ animals are hypersensitive to aldicarb, which is the opposite of the *unc-13* loss-of-function phenotype. This suggests that forcing microexon skipping in motor neurons does not merely result in a loss of UNC-13 activity, but confers new, perhaps overactive, UNC-13 function.

As a control, we also performed levamisole sensitivity assays, because aldicarb hypersensitivity can, in principle, be caused by either excessive presynaptic neuronal activity or by excessive post-synaptic receptor activity in muscles. Levamisole is an agonist for L-type acetylcholine receptors present on muscles, and thus reports on excitatory properties of post-synaptic muscles (Pir et al, 2016). Neither the $UNC$-$13_{included}$ nor the $UNC$-$13_{skipped}$ animals have a levamisole sensitivity phenotype (Fig. 5D). We also examined localization of post-synaptic markers in muscle, and found no difference in their localization (Fig. EV5A). Together, this indicates that *unc-13* microexon skipping causes presynaptic rather than post-synaptic neurotransmission defects.

To test the effect of the *unc-13* microexon on synaptic vesicle distribution, we examined the transgenic synaptic vesicle marker SNB-1::GFP. In order to visualize synapses from single presynaptic

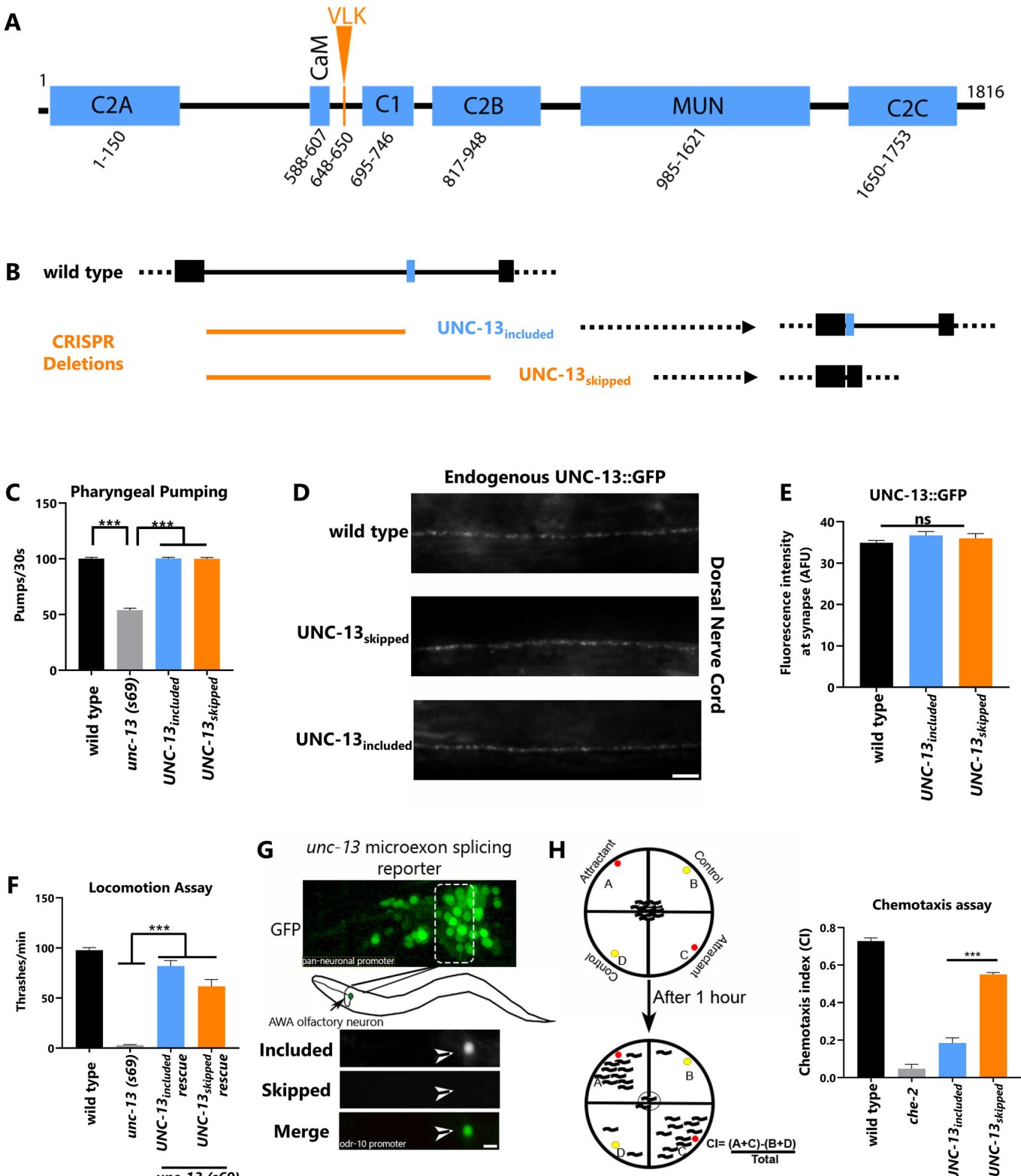

neurons, we imaged the PLM mechanosensory neuron, which, unlike ventral cord motor neurons, can easily be distinguished from neighboring neurites with single-cell resolution (Choudhary et al, 2017; Nonet, 1999). Like motor neurons, PLM neurons express only the *unc-13* included isoform (Fig. 5E). Loss of *unc-13* causes

increased SNB-1::GFP accumulation, as previously reported (Fig. 5F,G) (Richmond et al, 1999). Once again, forcing *UNC-13*<sub>skipped</sub> causes the opposite phenotype of the loss of function, resulting in decreased SNB-1::GFP accumulation (Fig. 5G). This reduced accumulation at the synapse is accompanied by increased

◄ **Figure 4.** *unc-13* **forced-isoform mutants reveal a requirement for microexon skipping in olfactory neurons.**

(A) Conserved UNC-13 protein domains and corresponding *C. elegans* amino acids #s. The *unc-13* microexon, encoding amino acids "VLK," is situated between CaM and C1 domains. (B) CRISPR-mediated genomic modifications (orange lines) to fuse the *unc-13* microexon to the upstream constitutive exon (*UNC-13*$_{included}$) or to remove the microexon entirely (*UNC-13*$_{skipped}$). (C) Pharyngeal pumping is reduced in *unc-13* (*s69*), a null allele, as previously reported (Kohn et al, 2000), but unaffected in either *UNC-13*$_{skipped}$ and *UNC-13*$_{included}$ strains. One-way ANOVA with multiple comparison analysis was performed. *p* values are displayed as $p_{adjusted}$ ($p_{adj}$) and shown as follows: ns (non-significant) = $p_{adj}$ > 0.05, *$p_{adj}$ < 0.05, **$p_{adj}$ < 0.01, ***$p_{adj}$ < 0.001. $p_{adj}$ value from L (left) to R (right): $p_{adj}$ < 0.0001, $p_{adj}$ < 0.0001. Error bars display the SEM. (D) Representative images of dorsal cord showing localization of endogenously GFP-tagged UNC-13. (E) Fluorescence quantification of the puncta in (D). One-way ANOVA with multiple comparison analysis was performed. ns = $p_{adj}$ > 0.05. Error bars display the SEM. (F) Locomotory behavior as assessed by thrashing assays. Overexpressing either the skipped or included *unc-13* isoform rescues the loss-of-function phenotype. One-way ANOVA with multiple comparison analysis was performed. $p_{adj}$ < 0.0001. Error bars display the SEM. (G) Upper panel, image of the nerve ring neurons, showing many neuronal cell bodies expressing the skipped version of *unc-13*. Many sensory neurons are situated in this region (dotted white rectangle). Middle panel: worm schematic showing the position of the AWA olfactory neuron marked by a black arrow. Bottom panel: representative image of *unc-13* microexon splicing reporter expressed in the AWA neuron via *odr-10* promoter as marked by white arrowhead. (H) Schematic of olfaction assay (left panel) used for pyrazine assay. The chemotaxis index (CI) is calculated as shown in the figure. Right panel: *UNC-13*$_{included}$ exhibits strong olfactory defects to 1 mM pyrazine. *che-2* (*e1033*) used as a chemotactic control (Bargmann et al, 1993). Unpaired *t*-test performed. *p* < 0.0001. Error bars display the SEM. Chemotaxis assay n = 3 biological replicates, 50–75 animals each. Other graphs, n = 10–20 animals. Scale bar = 10 μm. Source data are available online for this figure.

ectopic SNB-1::GFP in the minor neurite (Fig. 5H), which is not a presynaptic structure. Thus, the *unc-13* microexon is required for locomotory behavior, neuromuscular synaptic transmission, and synaptic vesicle localization.

## Inhibitory motor neurons are specifically sensitive to *unc-13* microexon skipping

Ventral cord motor neurons are composed of multiple neuron subtypes, and we next tested whether motor neuron subtypes have specific requirements for UNC-13 microexon inclusion. The aldicarb hypersensitivity paralysis phenotype in *UNC-13*$_{skipped}$ animals (Fig. 5B) could be caused by overactive excitatory neurons or by reduced compensatory activity of inhibitory neurons (Thapliyal et al, 2018). To distinguish between these possibilities, we took the *UNC-13*$_{skipped}$ strain and reintroduced microexon-included *unc-13* cDNA in specific cell types via transgenic overexpression. Surprisingly, expression in inhibitory neurons fully reverses the aldicarb hypersensitivity phenotype of the *UNC-13*$_{skipped}$ animals, while expression in excitatory neurons has no effect (Fig. 6A). Similarly, the locomotory deficits of *UNC-13*$_{skipped}$ animals are rescued by expression in inhibitory neurons, but not in excitatory neurons (Fig. 6B). These results indicate that inhibitory motor neurons are particularly affected by loss of *unc-13* microexon splicing.

Using these same transgenic lines, we tested whether over-expressing the *unc-13* skipped isoform in an otherwise wild-type background affects behavior. Indeed, overexpression in inhibitory neurons, but not in excitatory neurons, results in locomotory deficits in a subset of animals (Fig. 6C,D) as well as aldicarb hypersensitivity (Fig. 6E,F). These results suggest that the two UNC-13 isoforms might be in competition with each other, and overexpression of the inappropriate isoform (microexon skipped) can override the activity of the endogenously expressed isoform, thus acting in a dominant-negative manner in inhibitory neurons.

These cell-specific experiments suggest that insufficient GABA release from inhibitory motor neurons causes the locomotion and aldicarb phenotypes in *UNC-13*$_{skipped}$ animals. To further examine GABA signaling, we performed pentylenetetrazole (PTZ) pharmacological assays. PTZ is a $GABA_A$ receptor antagonist, and mutants with reduced GABA release are thus hypersensitive to PTZ, displaying head convulsions, full-body convulsions (tonic), or full paralysis (tonic-clonic) (Locke et al, 2008). Wild-type animals

expressing *unc-13* skipped cDNA in inhibitory neurons, but not excitatory neurons, display the characteristic reduced-GABA phenotypes (Fig. 6G). In fact, the PTZ sensitivity of these animals is as strong as our positive control *unc-25* mutants, which are defective in the GABA synthesis (Williams et al, 2004). These results demonstrate that aberrant expression of the UNC-13 microexon-skipped isoform in inhibitory neurons strongly reduces GABA synaptic transmission at the neuromuscular junction.

Having established that inhibitory neurons require the *unc-13* microexon-included isoform for proper GABA release, we next examined the organization of motor neuron synapses. We again visualized the synaptic vesicle marker SNB-1::GFP, this time driven in either excitatory or inhibitory motor neurons. *unc-13* loss-of-function mutation causes increased accumulation of SNB-1::GFP in both excitatory and inhibitory neurons (Fig. 6H), as previously reported (Richmond et al, 1999). However, overexpressing the microexon-skipped version of *unc-13* causes defects only in inhibitory motor neurons (Fig. 6H–J).

Taken together, these results demonstrate that *unc-13* microexon inclusion is crucial for motor neuron function, and that inhibitory GABAergic motor neurons are particularly sensitive to loss of microexon inclusion. GABAergic neurons overexpressing the skipped UNC-13 isoform cause neurotransmission and behavioral defects as well as altered clustering of synaptic vesicles at the presynaptic membrane.

## Broadly conserved features of MUN-domain gene microexon alternative splicing

Given the highly specific and precise regulation of the *unc-13* microexon, we asked whether the regulatory principles described here are unique to *unc-13*, or whether they represent a broader regulatory theme. We first focused on microexons encoded by other MUN-domain genes related to *unc-13*. In *C. elegans*, the MUN-domain genes *unc-13* and *unc-31* have been intensively studied, with *unc-13* playing a role in synaptic vesicle (neurotransmitter) release and *unc-31* in dense core vesicle (neuropeptide) release (Speese et al, 2007). Comparing MUN-domain genes across diverse phyla, it is notable that mammalian homologues of both *unc-13* (*Unc13a*, *Unc13b*, and *Unc13c*) and *unc-31* (*Cadps* and *Cadps2*) universally encode microexons (Fig. 7A). This suggests that microexons play important roles in MUN-domain gene function across evolution.

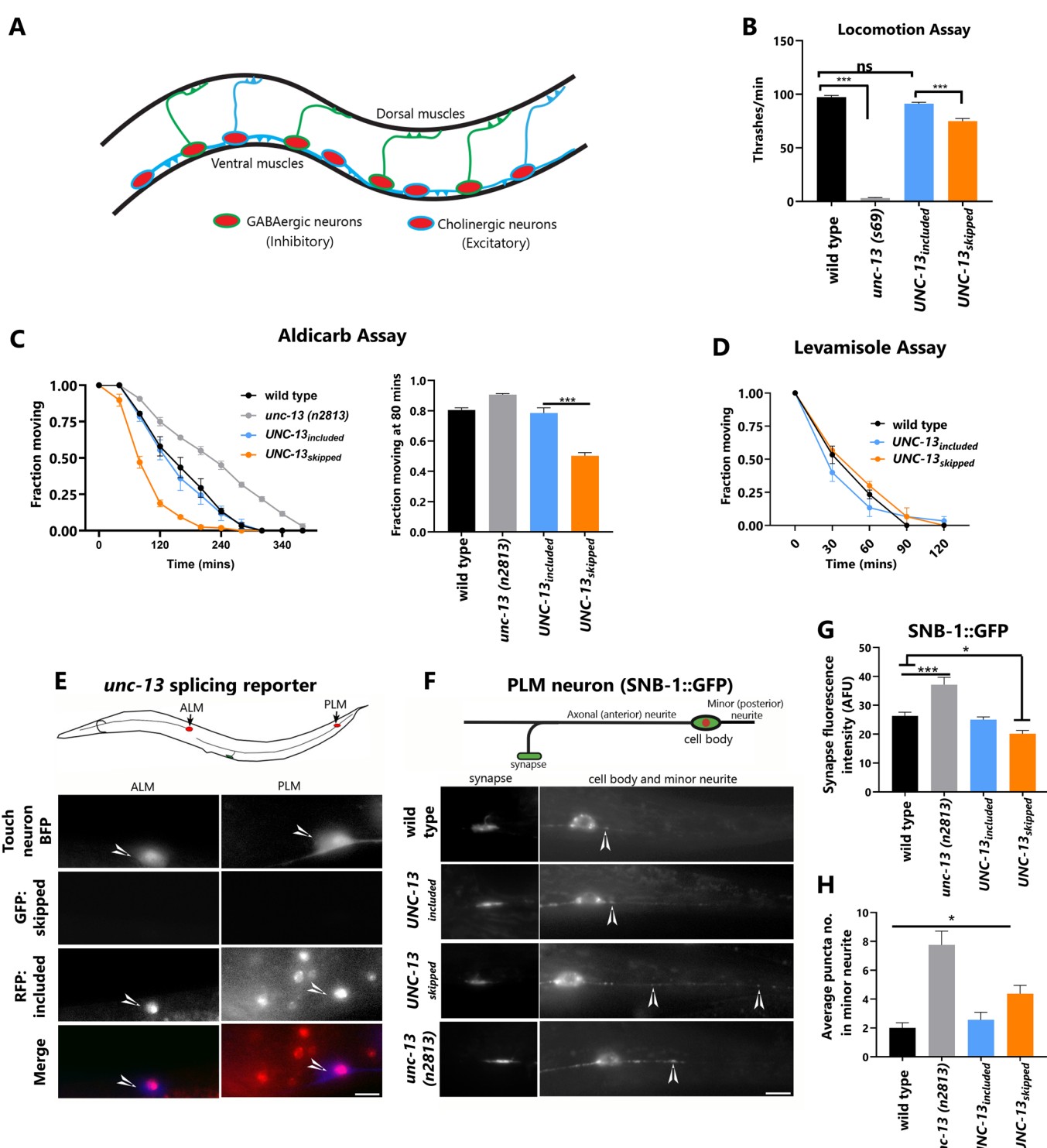

We first asked whether the other MUN-domain gene microexons have similar cell-specific splicing patterns as *unc-13*. Three such MUN-domain genes with microexons exist in *C. elegans*: *unc-13, unc-31*, and a more distantly related gene, *F54G2.1*. Neuron-specific RNA Seq shows that indeed all three microexons undergo similar cell-specific splicing patterns. For *unc-31*, the splicing pattern is nearly identical to *unc-13*, with microexon inclusion uniformly high in motor neurons and mechanosensory neurons, but low in chemosensory neurons. The *F54G2.1* microexon is also highly included in motor neurons and skipped in chemosensory neurons, but in this case, mechanosensory neurons exhibit microexon skipping (Fig. 7B). We constructed in vivo splicing reporters for both microexons, which agree with the RNA Seq data, for example, demonstrating strong microexon inclusion in motor neurons (Fig. 7C).

**Figure 5.** *unc-13* microexon inclusion is required for motor neuron function and synaptic vesicle localization.

(A) Cholinergic and GABAergic ventral cord motor neurons and their synaptic connections are made in the dorsal and ventral muscle. These drive the sinusoidal movement of the animal, orchestrated by alternate contraction and relaxation of the ventral and dorsal muscles driven by excitatory (cholinergic) and inhibitory (GABAergic) neurons. (B) Quantification of locomotion as determined by thrashing assays. One-way ANOVA with multiple comparison analysis was performed. $p_{adj}$ value from L to R: $p_{adj} < 0.0001$, ns = $p_{adj} = 0.0965$, $p_{adj} < 0.0001$. Error bars display the SEM. (C) Left panel, aldicarb response curve. $UNC-13_{skipped}$ animals are aldicarb hypersensitive, whereas *unc-13* (*n2813*) loss-of-function animals are aldicarb resistant, as previously reported (Kohn et al, 2000). Right panel, quantification of the fraction of animals moving at 80 min. Unpaired *t*-test performed. $p < 0.0001$. Error bars display the SEM. (D) Response curve for levamisole assay. No significant difference was found among genotypes. (E) Upper panel: Schematic of worm showing position of PLM and ALM mechanosensory neurons by black arrow. Lower panel: Representative images of animals with mechanosensory neurons marked by *mec-7p::BFP* reporter, co-expressing *unc-13* splicing reporter. The white arrowhead marks the ALM and PLM neurons. Mechanosensory neurons exclusively express microexons, including the *unc-13* isoform. (F) Upper panel: schematic of PLM neuron cell body, long neurite (axonal neurite) with synapses, and minor neurite ending in the tail region. Lower Panel: representative images of SNB-1::GFP localization at the presynaptic region (left panel) and minor neurite (right panel). White arrowheads mark SNB-1::GFP puncta in minor neurites. (G, H) Quantification of the intensity of SNB-1::GFP cluster at presynaptic region (G) and puncta number in the minor neurite (H). One-way ANOVA with multiple comparison analysis was performed. $p_{adj}$ value from L to R for (G): $p_{adj} < 0.0001$, $p_{adj} = 0.02$ and for (H): $p_{adj} = 0.02$. Error bars display the SEM. $n = 15$–20 animals. Scale bar 10 µm. Source data are available online for this figure.

Next, we tested whether cell-specific splicing patterns of MUN-domain-gene microexons are unique to this gene family, or are global attributes of microexon splicing. Focusing on chemosensory and motor neurons, we find a minor trend toward lower microexon inclusion in chemosensory neurons and higher inclusion in motor neurons. However, the MUN-domain-gene microexons are outliers, representing some of the most highly included microexons in motor neurons, and the most highly skipped microexons in chemosensory neurons (Fig. 7D). Therefore, we conclude that the striking pattern of cell-specific microexon splicing described here is unique to this small subset of microexons, while other microexons and subsets of microexons likely have their own unique cell-specific splicing patterns.

Since all three MUN-domain-gene microexons share similar cell-specific splicing profiles, we next asked whether they are all co-regulated by EXC-7 and MBL-1. Indeed, RNA Seq reveals that all three are modestly affected in *exc-7* or *mbl-1* mutants, but are completely skipped in *exc-7; mbl-1* double mutants, which is also supported by the RT-PCR analysis (Fig. 7E,F). This is not a universal feature of microexons, as only a small subset of microexons are dysregulated in *exc-7; mbl-1* double mutants. Indeed, among microexons with splicing dysregulation that is more extreme in double mutants than predicted by simple addition of the single mutant effects, three of the top four are in MUN-domain genes (Fig. 7G). Therefore, we conclude that EXC-7 and MBL-1 specifically co-regulate a small network of microexons encoded by MUN-domain synaptic transmission genes.

Because *exc-7; mbl-1* double mutants completely lack *unc-13* microexon inclusion, we wondered if these mutants might have synaptic transmission phenotypes similar to $UNC-13_{skipped}$ mutants. Indeed, *exc-7; mbl-1* double mutants are hypersensitive to aldicarb (Fig. 7H). We then tested whether this synaptic transmission defect is due to loss of *unc-13* microexon inclusion by constructing a triple mutant *exc-7; mbl-1; $UNC-13_{included}$*. Strikingly, this restored the synaptic transmission phenotype nearly back to wild-type levels (Fig. 7H). Thus, although EXC-7 and MBL-1 control splicing and expression of hundreds of genes (Napier-Jameson et al, 2024), aberrant splicing of a single microexon is responsible for these synaptic transmission defects.

Finally, we asked whether the cell-specific splicing of MUN-domain-gene microexons might be a universal feature across species. To address this, we turned to available cell-specific bulk RNA Seq data from mouse olfactory neurons and mouse motor neurons (Bandyopadhyay et al, 2013; Kanageswaran et al, 2015).

We found only one mammalian MUN-domain gene, *Cadps*, to be covered with sufficient depth to analyze microexon splicing. This gene, homologous to *C. elegans unc-31*, contains a 9 nt exon with an amino acid sequence (GLK) similar to the *C. elegans unc-13* sequence (VLK) (Fig. 7I). The cell-specific RNA Seq data reveal that indeed the microexon is highly included in motor neurons, and highly skipped in olfactory neurons, much as in *C. elegans* (Fig. 7I). This finding is consistent across biological replicates (Fig. 7J), and suggests the cell-specific microexon splicing patterns described here might represent broadly-applicable regulatory regimes across diverse nervous systems.

## Discussion

Here, we identify widespread alternative splicing of microexons across cell types within the nervous system. Not only are microexons preferentially included within the nervous system compared to other tissues, but within the nervous system, additional heterogeneity exists across neuronal cell types. The alternative microexon splicing reported here is likely to be a substantial underestimate, given that we only focused on genes that are broadly expressed across neuronal cell types. We anticipate additional cell-specific alternative microexon splicing in genes expressed in more restricted patterns. Our observations dovetail nicely with recent global brain-region-specific and cell-specific transcriptomic analyses suggesting the existence of substantial alternative microexon splicing across neuronal cell types (Ciampi et al, 2022; Han et al, 2024; Parada et al, 2021).

We also demonstrate molecular mechanisms by which such microexon alternative splicing can be established. In the case of MUN-domain microexons, exemplified by *unc-13*, microexon inclusion is mediated by the RBPs EXC-7 and MBL-1, for which consensus *cis* elements reside in the downstream intron. We propose that binding of these two RBPs recruits PRP-40, a component of the U1 snRNP. We previously showed that PRP-40 facilitates microexon splicing via intron definition, thus facilitating splicing of microexons that would be inefficiently spliced via typical exon definition mechanisms (Choudhary et al, 2021). Therefore, our data suggest MUN-domain microexon alternative splicing is governed by cell-specific EXC-7 and MBL-1 expression: cells with low expression skip the microexon due to inefficient recruitment of PRP-40, while cells with high expression efficiently recruit PRP-40 and include the microexon. This model is supported by single-cell

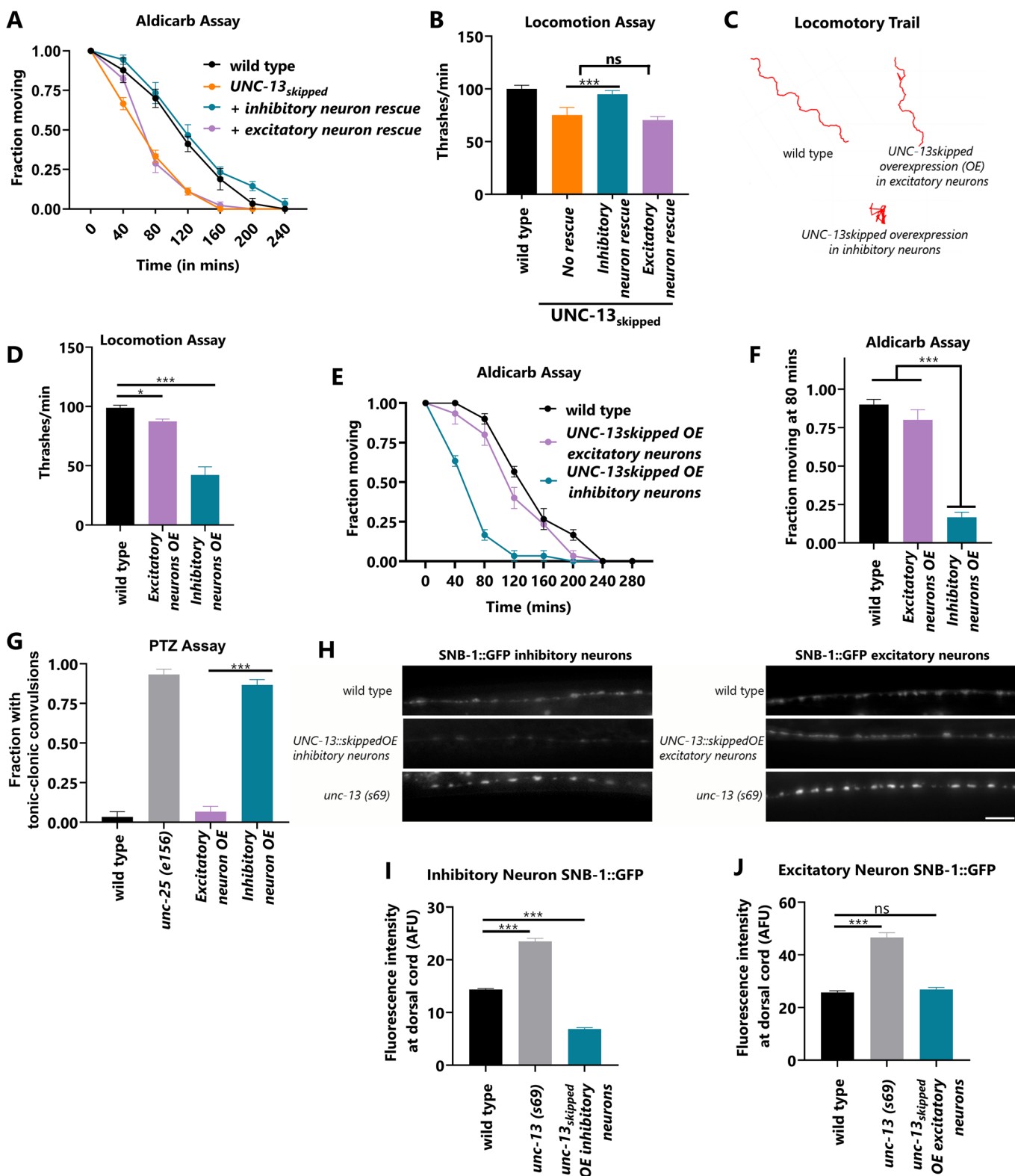

expression data (Liska et al, 2023) and translational-fusion fosmid reporter transgenes (Thompson et al, 2019), indicating that olfactory neurons exhibit low or undetectable levels of both RBPs (Fig. EV5), while motor neurons express both EXC-7 and MBL-1 (cholinergic neurons) or MBL-1 (GABAergic neurons) (Fig. EV5).

With respect to microexon function, previous experiments have identified microexons whose splicing is required for neuronal development and function (Gonatopoulos-Pournatzis et al, 2020; Johnson et al, 2019; Lin et al, 2020; Nakano et al, 2018). One relevant example is a 6 nt exon in mouse Unc13b

**Figure 6. Inhibitory motor neurons are particularly sensitive to loss of *unc-13* microexon inclusion.**

(A) Aldicarb response curve for UNC-13$_{skipped}$ and for cell-specific cDNA overexpression (OE) of UNC-13 included in the *UNC-13*$_{skipped}$ background. Expression in GABAergic neurons restores aldicarb sensitivity. (B) Thrashing assay reveals rescue of *UNC-13*$_{skipped}$ by expression of UNC-13 included cDNA in GABAergic neurons. One-way ANOVA with multiple comparison analysis was performed. $p_{adj}$ value from L to R: $p_{adj} < 0.0001$, ns = $p_{adj} = 0.38$. Error bars display the SEM. (C) Representative worm tracks generated for UNC-13skipped cDNA OE in wild-type worms. OE in inhibitory neurons causes a range of locomotory phenotypes, from mild to strong (displayed track is representative of strong phenotype). OE in excitatory neurons shows no such phenotypes. (D) Thrashing quantification of UNC-13skipped cDNA OE in a wild-type background. One-way ANOVA with multiple comparison analysis was performed. $p_{adj}$ value from L to R: $p_{adj} = 0.012$, $p_{adj} < 0.0001$. Error bars display the SEM. (E) Aldicarb response curve for cell-specific OE of UNC-13skipped cDNA. (F) Quantification of the percentage of moving animals from (E) moving at 80 min. One-way ANOVA with multiple comparison analysis was performed. $p_{adj} < 0.0001$. Error bars display the SEM. (G) Quantification of tonic-clonic convulsion behavior at 45 mins for PTZ assay. Wild type and *unc-25 (e156)* act as negative and positive controls, respectively. For this assay, mild to strong *unc* animals were selected for UNC-13skipped OE in inhibitory neurons. One-way ANOVA with multiple comparison analysis was performed. $p_{adj} < 0.0001$. Error bars display the SEM. (H) Representative images of SNB-1::GFP expressed in the dorsal cord of inhibitory (left panel) and excitatory (right panel) neurons. SNB-1::GFP puncta are reduced in mild to strong *unc* animals with UNC-13skipped OE in inhibitory neurons. Scale bar 10 μm. (I, J) Quantification of SNB-1::GFP puncta intensity in the dorsal cord of GABAergic (I) and cholinergic (J) nervous system. One-way ANOVA with multiple comparison analysis was performed. $p_{adj}$ value from L to R for (I): $p_{adj} < 0.0001$, $p_{adj} < 0.0001$ and for (J): $p_{adj} < 0.0001$, ns = $p_{adj} = 0.07$. Error bars display the SEM. $n = 15–20$ animals. Source data are available online for this figure.

(Quesnel-Vallieres et al, 2015), which appears to be evolutionarily unrelated to the 9 nt microexon we studied. The 6 nt exon in Unc13b was shown to be necessary for proper outgrowth of cultured mouse cortical neurons (Quesnel-Vallieres et al, 2015). It will be interesting to determine whether this microexon also undergoes alternative splicing across neuron types, and whether the exon-skipped isoform performs an essential function as well.

Here we show that for a 9 nt microexon in the *unc-13* gene, both the included and skipped isoforms are functionally required, depending on the neuron type. We thus extend the concept of neural microexons essential for nervous system function to also include neural microexons whose inclusion *and* skipping are essential, depending on the neuron type. Alternative inclusion of a mere three amino acids in UNC-13 has a major effect on neuronal function, and we speculate that they function as modulators of synaptic transmission rate. Indeed, the microexon is located in a linker region between the regulatory CaM and C1 domains of UNC-13. Our pharmacological experiments indicate that preventing microexon inclusion causes insufficient GABAergic neurotransmission. Thus, the microexon-skipped version of UNC-13 might represent an isoform with lower synaptic transmission activity, while the microexon-included version represents a higher-activity isoform. This could be a subtle but elegant way to tune the strength of synaptic transmission according to cell-type-specific functional needs.

Genes in the MUN-domain family invariably encode microexons across wide evolutionary distances (worm, fly, mouse, and human). Moreover, these microexons share regulatory properties: first, in *C. elegans* they are all co-regulated by *exc-7* and *mbl-1*. Second, their single-neuron splicing properties in *C. elegans* are similar (e.g., included in motor neurons, skipped in olfactory neurons). Third, this single-neuron splicing pattern holds in the mouse for the *Cadps* microexon (the only family member with sufficient data to test). This suggests that there might be conserved microexon "regulatory modules" with shared properties of cell-specific expression, regulation, and function. We note that besides MUN-domain genes, very few microexons are co-regulated by *exc-7* and *mbl-1*; nor is there a global pattern of microexon inclusion in motor neurons and skipping in olfactory neurons. Rather, we propose that each microexon regulatory module is subject to its own cell-type-specific expression and regulatory regime. For example, microexons in cell adhesion molecules or ion channels might be subject to orthogonal splicing patterns, thus further diversifying and tuning neuronal development and excitability.

# Methods

## Reagents and tools table

| Antibodies | Company | Catalog |
|---|---|---|
| Anti-flag trap | Chromotek/Proteintech | ffa |
| Anti-flag | Sigma-Aldrich | F3165 |
| Anti-mScarlet | Chromotek/Proteintech | 6G6 |
| Anti-mouse secondary | CST | 7076 |
| **Chemicals** | | |
| Pentazylenetetrazole | TCI America | P00465G |
| Aldicarb | Millipore Sigma | 33386-100MG |
| Levamisole | Thermo Scientific Chemicals | AC187870100 |
| **Primers used in this study** | **Forward primer** | **Reverse primer** |
| unc-13 microexon | 5'AACGAAGACAAGTATTCCACT3' | 5'ATTCGCCAGACCTGCTT3' |
| unc-31 microexon | 5'ATCCGCGTGACGGAATTCTG3' | 5'GATCTCGGAATGGTCGGAAAG3' |
| F54G2.1 | 5'GAAACTTGTAAAAATTCTCTCG3' | 5'TTATAATATGCTCTGACGTTGAG3' |

## Worm maintenance and genetics

All *C. elegans* strains were cultured at 20 °C as described. Bristol N2 was used as the wild-type strain. The integrated transgenes and mutants used in this study are: *adnIs4, adnIs5, nuIs158, kyIs158, nuIs299, nuIs283, jsIs37, juIs76, ItSi914, exc-7 (rh252), mbl-1 (wy560), unc-75(csb20), exc-7(csb28), prp-40 (csb3), asd-1(csb32), uaf-1 (n4588), fust-1 (csb22), fox-1 (csb39), mec-8 (csb23), tdp-1 (csb37), che-2 (e1033), unc-13 (n2813), unc-13 (s69)*. CRISPR edited strains: UNC-13$_{included}$/*unc-13 (syb2985)*, UNC-13$_{skipped}$/*unc-13 (syb2986)*, UNC-13::GFP/ *unc-13(syb5767)*, UNC-13$_{included}$::GFP/ *unc-13(syb5822)*, UNC-13$_{skipped}$::GFP/ *unc-13 (syb5875)*, MBL-1::3x-FLAG/ PHX5263 *mbl-1(syb5263)*, prp-40::mScarlet/ *prp-40 (syb1811), ot970*. Following extrachromosomal transgenic strains were used where 20 ng/μl of constructs were injected: ADN299 = *Ex [prp-40$_{OE}$; pfox-1::BFP]*, ADN932 = *Ex [pmec-2::BFP],*

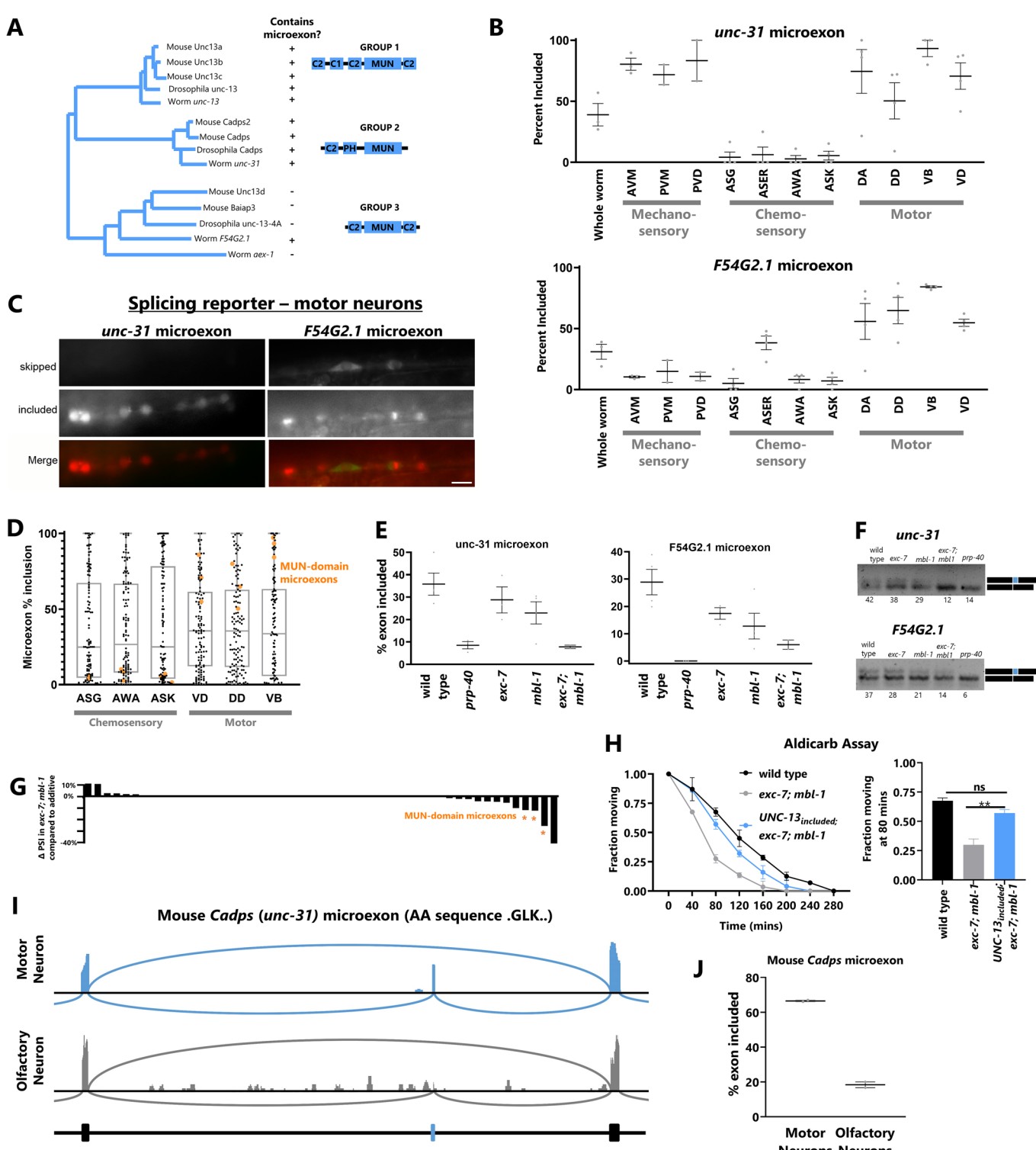

ADN1257 = Ex [pODR-10::UNC-13WTWASR; punc-122::GFP], ADN1054 = Ex [prgef::UNC-13WASR] with upstream intron deletion; ADN1049 = Ex [prgef::UNC-13WASR] with downstream intron deletion; ADN1484 = Ex[punc-25:: UNC-13skipped; punc-122::GFP], ADN1480 = Ex[punc-17::UNC-13skipped;pmyo-2::RFP], ADN1376 = Ex[punc-25:: UNC-13included; punc-122::RFP], ADN1372 = Ex[punc-17::UNC-13included; punc-122::GFP], ADN1346 = Ex [prgef::UNC-13included;punc122::RFP], ADN1347 = Ex [prgef::UNC-13skipped; pmyo3::BFP], ADN1217 = Ex [prgef::UNC-13WASR] with EXC-7 motif mutations, ADN1231 = Ex [prgef::UNC-13WASR] with MBL-1 motif mutations, ADN1230 = Ex [prgef::UNC-13WASR] with EXC-7 and MBL-1 motif mutations ADN1565 = Ex [prgef::UNC-31WASR], ADN1561 = Ex [prgef::F54G2.1WASR], ADN 1563 = Ex [punc-25::BFP], ADN 228 = Ex [MBL-1::RFP fosmid + EXC-7::GFP fosmid +

**Figure 7. Broadly conserved features of MUN-domain gene microexon alternative splicing.**

(A) Phylogenetic tree of MUN family members based on Pei et al, 2009 (Pei et al, 2009), revealing three clusters: UNC-13 family, UNC-31/CAPS, and UNC-13d (left panel). Presence of alternatively spliced microexon marked as "+" and its absence as "−". Note that all Group 1 and 2 genes encode microexons in worms, flies and mice. (B) % inclusion for microexons in *unc-31* (upper panel) and *F54G2.1* (lower panel) in different neuronal subtypes. Both microexons are highly included in motor neurons. Error bars display the SEM. (C) Splicing reporters for *unc-31* and *F54G2.1* microexons in ventral cord motor neurons. (D) % inclusion values for all detectable microexons in three representative chemosensory neurons and three representative motor neurons. Median spliced-in values (central line) are modestly higher in motor neurons than in olfactory neurons, but MUN-domain microexons (orange dots) are extreme outliers of skipping in olfactory neurons and inclusion in motor neurons (boxes represent 25th and 75th percentiles). The plot displays whiskers at the maxima and minima (0 and 100). (E, F) % inclusion values obtained from RNA seq data (E) and further verified by semi-quantitative RT-PCR (F) for MUN-domain microexons reveal co-regulation by EXC-7 and MBL-1. Numbers below bands indicate PSI values determined by gel densitometry. Error bars display the SEM. (G) ΔPSI values for observed *exc-7; mbl-1* double mutants compared to expected ΔPSI values based on additive effects of single mutant *exc-7* and *mbl-1* ΔPSIs. Thus, negative, and positive values represent microexons that are more skipped or more included, respectively, compared to an additive-effect model. Orange asterisks highlight microexons in MUN-domain family genes. (H) Left panel: aldicarb response curve, right panel: quantification of the fraction of animals moving at 80 min. *exc-7; mbl-1* double mutants are aldicarb hypersensitive, and this phenotype is rescued by the addition of an UNC-13$_{included}$ mutation. One-way ANOVA with multiple comparison analysis was performed. $p_{adj}$ value from L to R: ns = $p_{adj}$ = 0.07, $p_{adj}$ < 0.01. Error bars display the SEM. (I) Sashimi plot for alternatively spliced CADPS/*unc-31* microexon in mouse motor and olfactory neurons. (J) PSI value for the *unc-31/Cadps* alternative microexon in the mouse motor and olfactory neurons as calculated from RNA seq data. $n$ = 20 animals. Scale Bar 10 µm. Source data are available online for this figure.

*punc-17::BFP].* WASR = two-color splicing reporters. Genome-edited *syb* alleles were generated by SunyBiotech. Lab strains are available on request.

## Plasmids construction

*C. elegans* plasmid insert constructs were synthesized de novo (Invitrogen) and promoter::insert fusions were generated by using Gateway recombination with various previously-generated promoter destination vectors, as described previously (Reece-Hoyes and Walhout, 2018).

## Behavioral assay

For the locomotory assay, larval stage 4 (L4) worms of respective genotypes were selected and put in a drop of M9 buffer, and the number of thrashes was counted for 1 min (unblinded assay). For the chemotaxis assay, L4 stage animals (50–100) were put in the center of the unseeded plate, and in the opposite quadrants, either the attractant as pyrazine (1 mM) or an ethanol drop was kept. After 1 h of incubation, the animals were scored in different quadrants and the chemotaxis index (CI) was calculated as follows: (Worms in both the test quadrants-worms in the control quadrants)/ Total number of the worms in all the quadrants (also refer Fig. 4H). Each set of experiments was done in triplicate and performed at least thrice.

## Pharmacological assays

### Aldicarb assay

From a 100 mM aldicarb stock solution (prepared in ethanol), 0.4 mM NGM fresh plates were made and seeded with OP50 E. Coli. Around 20 worms (L4 stage) from each genotype were kept on the plates (in triplicates), and scored for the paralysis in 40 min intervals. Animals were considered paralyzed if no movements/ body bending were observed by touching the head and tail (Mahoney et al, 2006).

### Levamisole assay

The levamisole plates were prepared in a similar way as aldicarb assay, but with a concentration of 0.5 mM, and the paralysis was scored at the 30 min interval as described previously [22].

Pentazylenetetrazole (PTZ) assay: For the PTZ assay, a 100 mg/ml stock solution was made, and plates were prepared freshly with a concentration of 10 mg/ml. Plates were allowed to be dried and seeded with concentrated OP50 E. coli same day. The experiments were performed on the same day in triplicates. Our scoring was based on the tonic and tonic-clonic convulsion phenotype across the worms, as described previously by Locke et al, 2008 (Locke et al, 2008).

## Imaging experiments

All the imaging experiments were performed on a Zeiss AxioImager Z1 with a 63X oil immersion objective and in some instances a 20X air objective. Images were processed with Image J. For fluorescence quantification, images across various genotypes were acquired at constant exposure and gain. The quantification is done by Image J.

## RNA extraction and semi-quantitative RT-PCR analysis

Whole animal RNA extraction was done with Direct-zol RNA Miniprep kits (Zymo Research R2050). The semi-quantitative RT-PCR was done as previously described (Choudhary et al, 2021). The RT-PCR products were analyzed on agarose gel electrophoresis and the quantification is done on Image J.

## RNA seq and data analysis

For interrogation of splicing in mutant RBP backgrounds, we analyzed existing RNA Seq data generated from whole L4 animals (Liang et al, 2022; Napier-Jameson et al, 2024; Norris et al, 2014; Taylor et al, 2023). Analysis of microexon alternative splicing was performed using CeNGEN FACS-sorted neuron-specific deep transcriptomes (Wolfe et al, 2025). For display in Fig. 1, only microexons whose genes are expressed in all neuron types in the CeNGEN database are presented. Analysis of mouse microexon splicing took advantage of FACS sorting data obtained in (Bandyopadhyay et al, 2013; Kanageswaran et al, 2015).

## Immunoprecipitation and western blot

Mixed stage animals were collected and homogenized with RIPA buffer supplemented with protease inhibitors (rtak Proteintech, Protease inhibitor Roche). The homogenized lysate was centrifuged at 8000 × g and the supernatant was collected. The supernatant was incubated with Anti-Flag antibody attached to agarose beads

(Proteintech), and various immunoprecipitation steps were done as described in rtak catalog Proteintech, except the incubation time for flag-tagged agarose beads with homogenate was performed for 4–5 h. The resultant output was probed by western blot, and the following antibodies were used: Mouse Anti-RFP (Proteintech), Mouse Anti-Flag (Sigma), Anti-Mouse-HRP (CST).

## Statistical analysis

Statistical analysis was performed with GraphPad Prism 8. Statistical tests used as appropriate were Student *t*-test, one-way ANOVA with Bonferroni multiple comparison. All the data shown are mean ± standard error of the mean. For transcriptome-wide analyses, differential microexon inclusion was determined by JUM (Wang and Rio, 2018) using Bonferroni-adjustment for multiple-hypothesis testing-adjusted *p* values.

# Data availability

This study includes no data deposited in external repositories.

The source data of this paper are collected in the following database record: biostudies:S-SCDT-10_1038-S44319-025-00493-7.

# Peer review information

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

## Acknowledgements

We are grateful for the following funding support: National Institute of General Medical Sciences of the National Institutes of Health [R35GM133461]; National Institute of Neurological Disorders and Stroke of the National Institutes of Health [R01NS111055]. We thank the Norris Lab, and Canyon Calovich-Benne in particular, for critical reading of the manuscript.

## Author contributions

**Bikash Choudhary**: Conceptualization; Data curation; Investigation; Methodology; Writing—original draft; Writing—review and editing. **Rebekah Napier-Jameson**: Investigation; Methodology. **Adam Norris**: Funding acquisition; Writing—original draft; Project administration; Writing—review and editing.

Source data underlying figure panels in this paper may have individual authorship assigned. Where available, figure panel/source data authorship is listed in the following database record: biostudies:S-SCDT-10_1038-S44319-025-00493-7.

## Disclosure and competing interests statement

The authors declare no competing interests.

# Expanded View Figures

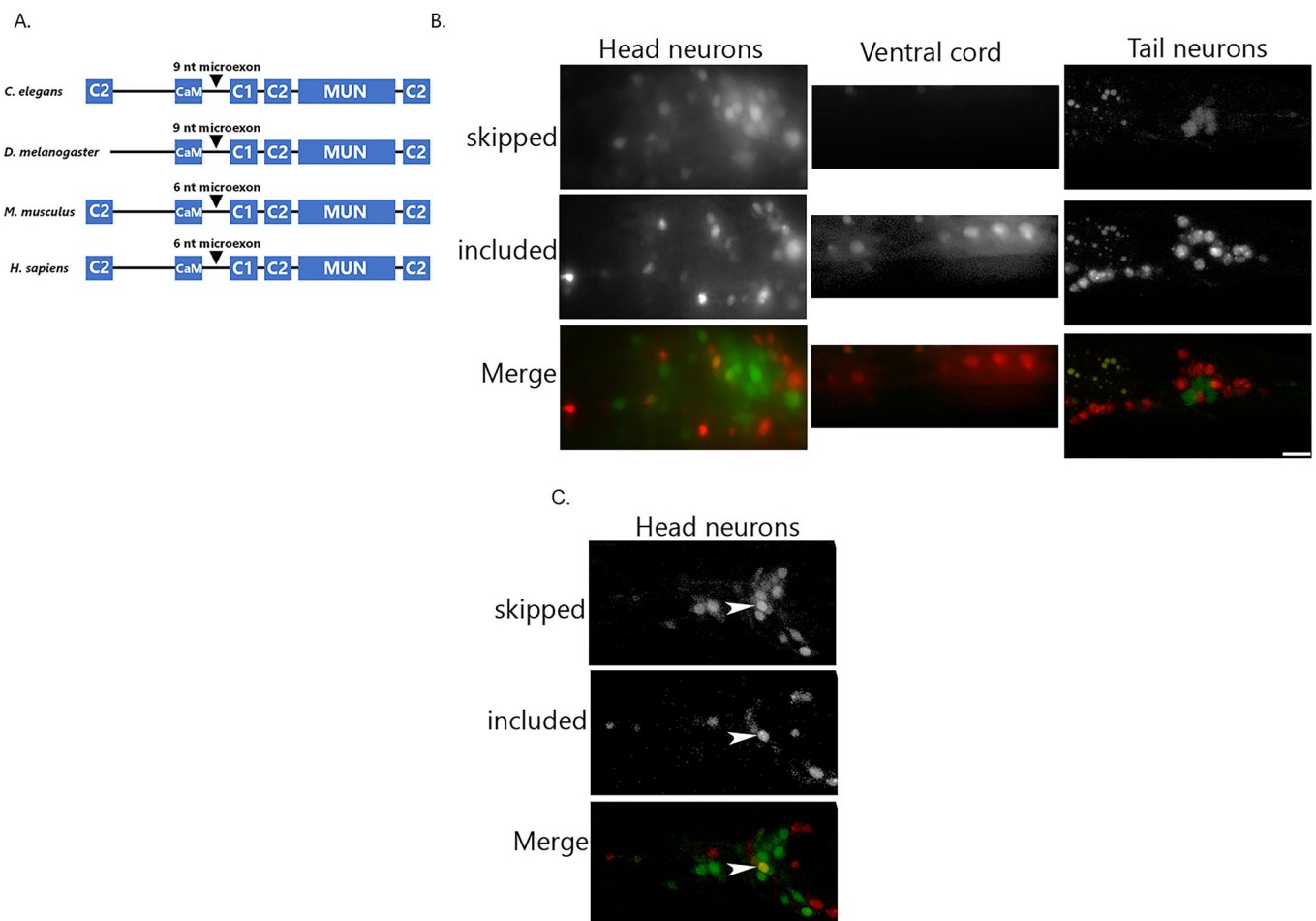

**Figure EV1.  The presence of an alternative microexon is conserved across species and its splicing pattern is invariant across multiple isolated independent integrated transgenic lines in *C. elegans*.**

(A) Schematic of the position of the alternatively spliced microexon in the *unc-13* transcript across various species. In mice and humans, it's 6 nt microexon, whereas in *C. elegans* and *Drosophila*, it's 9 nt. Black arrowheads mark the position of the microexon in different species. (B) *unc-13* microexon-splicing pattern in various regions of the nervous system of an independently isolated integrated transgenic line. (C) A neuron marked by a white arrowhead expressing both skipped and included versions in the nerve ring neurons of the *unc-13* microexon splicing reporter expressing animal. Scale bar 20 μm.

A.

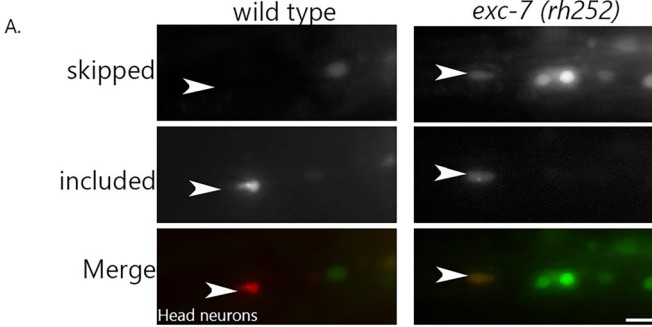

**Figure EV2. Alternative splicing of the *unc-13* microexon is affected in a neuron-specific manner in *exc-7* animals.**

(A) A region of the head neurons, where an identified neuron in wild type exclusively has RFP, whereas in *exc-7(rh252)* it has both, GFP and RFP signal. The white arrowhead marks that neuron. Scale bar 10 μm.

A

cis motif EXC-7
mutated

wild type

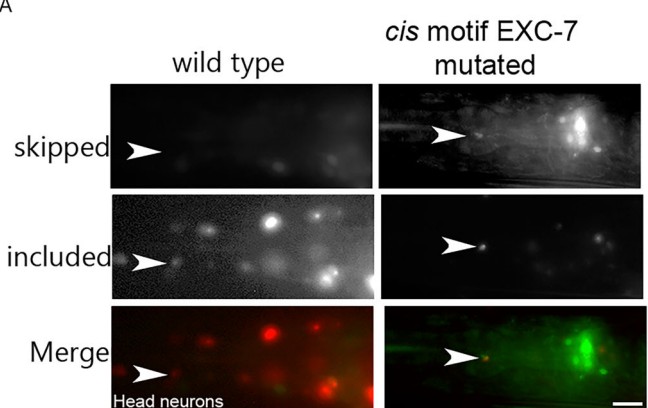

skipped

included

Merge

Head neurons

B

*cis* motif EXC-7 and MBL-1
mutated

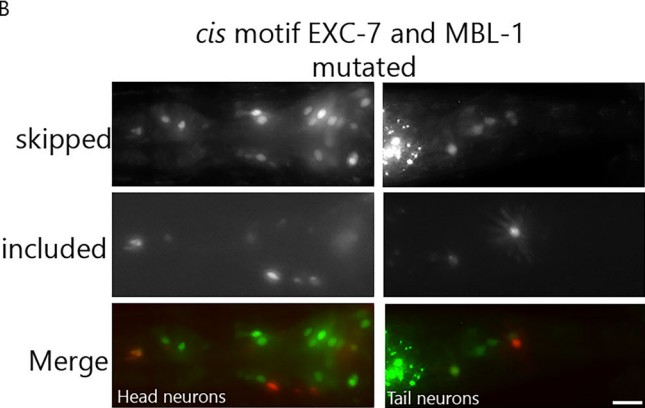

skipped

included

Merge

Head neurons   Tail neurons

**Figure EV3.   Mutating *cis* motifs for EXC-7 and/or MBL-1 affect splicing in a neuronal-subtype-specific manner.**

(**A**) A neuron in pharyngeal region shows the expression of GFP and RFP, whereas in wild type it expresses only included form. The white arrowhead marks that individual neuron. (**B**) Representative image showing *cis* motif mutation in both EXC-7 and MBL-1, where a set of animals (~20% of animals) has lesser RFP neuronal cell bodies in head and tail neurons compared to wild type. Scale bar 20 μm.

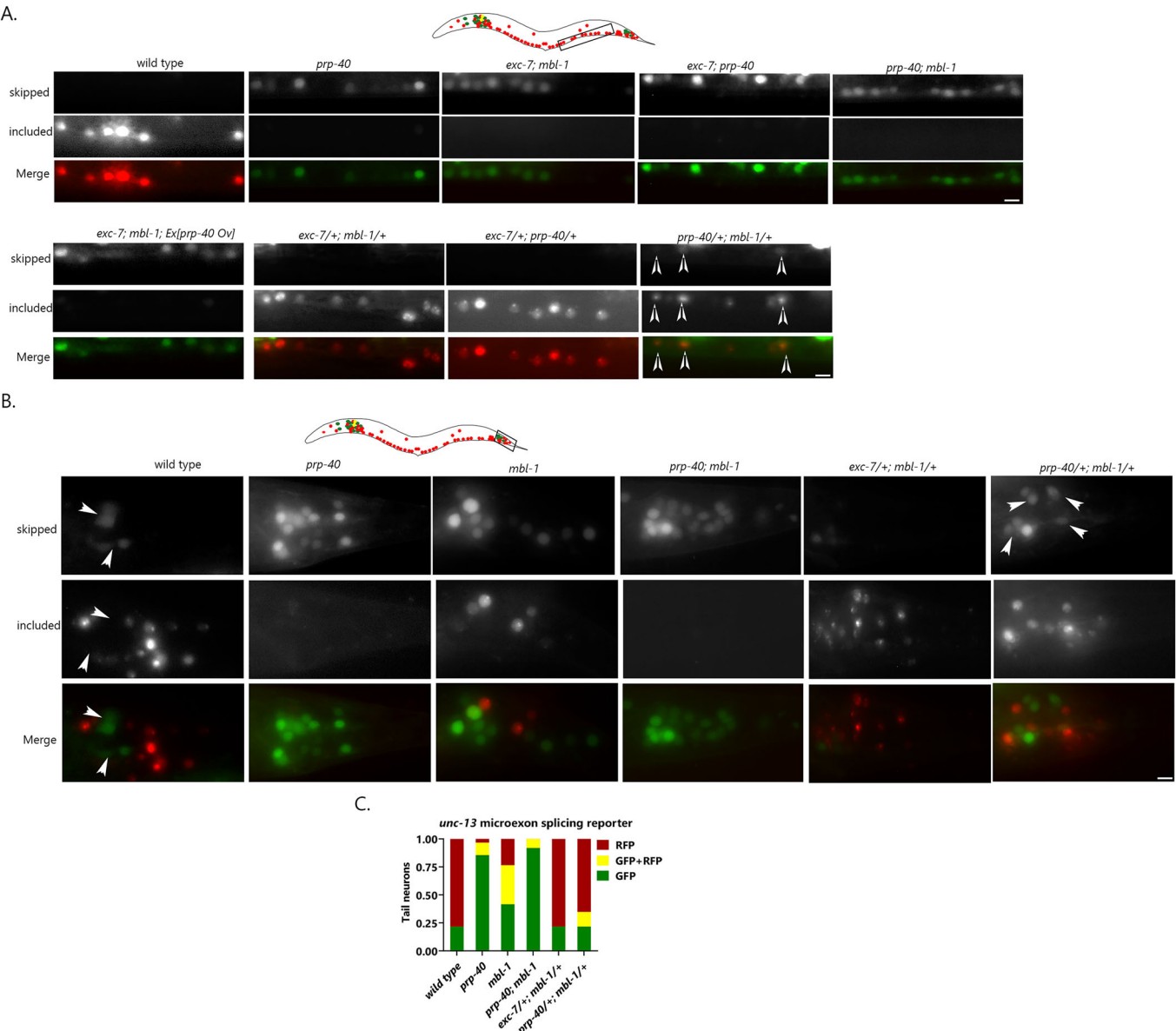

**Figure EV4. Genetic interaction between splicing regulators for the alternative splicing of *unc-13* microexon in different neuronal subtypes (also see Fig. 3G, H in the main section for quantification).**

(A) Upper panel: Worm schematic showing the splicing reporter pattern and the below images are from the area marked by dotted rectangle in the ventral cord region. Note that the wild-type control panel is the same worm as in Figs. 2D and 1F. Bottom panel: Representative images of the ventral cord region of the for various genotypes. White arrowheads mark the GFP appearance in the double heterozygous animals. (B) Upper panel: A worm schematic showing splicing reporter with the area marked by a dotted rectangle in the tail region is represented in the below mentioned images for various genotypes. Bottom panel: Representative images of the tail neurons in the mentioned genotypes. White arrowheads in the wild type panel marks a set of tail neurons expressing the skipped isoform (GFP) (C) Quantification of the splicing analysis of the tail neurons in the indicated genotypes. $n = 15$–20 animals. Scale bar 10 μm.

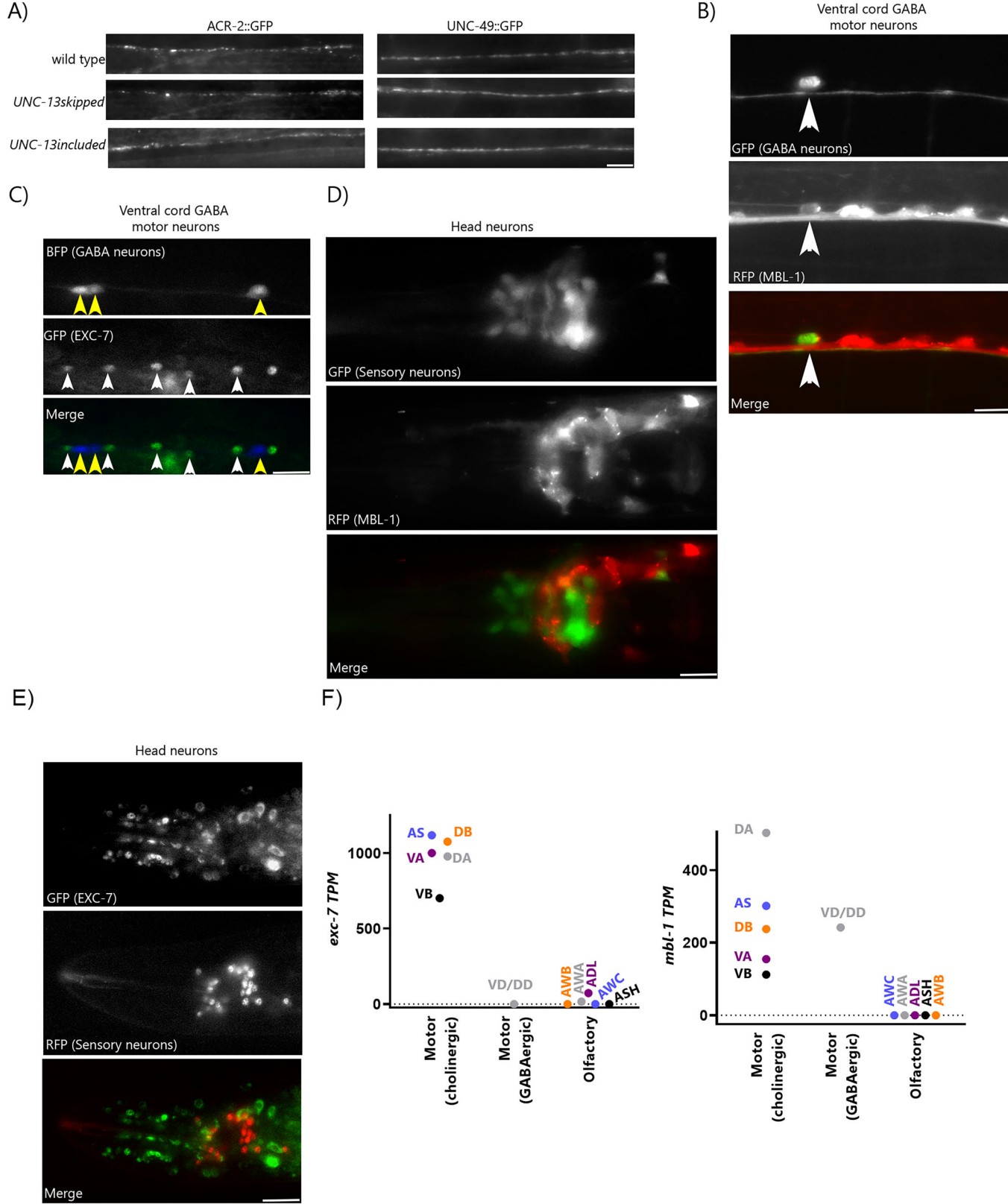

◀ **Figure EV5. Localization of SNB-1::GFP in dorsal cords and expression analysis of RBPs EXC-7 and MBL-1 by using reporter strains in addition to the CeNGEN database.**

(A) Representative images of localization of post-synaptic markers such as acetylcholine receptors (left panel) and GABA receptors (right panel) in the muscle compartment. (B) The upper panel shows GABA motor neurons (white arrowhead) in the ventral cord, which colocalize with MBL-1::RFP driven by a fosmid. (C) GABA motor neurons in the ventral cord marked by BFP (yellow arrowheads) driven under *unc-25* promoter does not colocalize with EXC-7::GFP (marked by white arrowheads) in ventral cord neurons. (D) Sensory neurons marked by GFP driven under *osm-6* promoter and MBL-1::RFP expression by the fosmid, indicates many of the sensory neurons does not have detectable MBL-1 expression. (E) The upper panel indicates EXC-7::GFP expression in the head region, and the sensory neurons marked by HIS-11::mcherry driven by *osm-6* promoter (middle panel). Sensory neurons have little to no detectable expression of EXC-7 as scored by the reporter analysis. (F) Single-cell RNA seq analysis for *exc-7* and *mbl-1* transcripts in multiple sensory and motor neurons (GABA and cholinergic neurons). Analysis indicates significant transcript level in motor neurons for *mbl-1*, whereas a set of sensory neurons lack the same, whereas exc-7 transcripts are lower or undetectable in both sensory and GABA motor neurons, but enriched in cholinergic motor neurons. Scale bar 20 μm.

 