## [Peer Review File · EMBO Reports]

Regulated microexon alternative splicing in single neurons tunes synaptic function

Bikash Choudhary, Rebekah Napier-Jameson, and Adam D Norris

Corresponding author(s): Adam D Norris (adamn@ucr.edu)

Review Timeline:

Submission Date:	17th Jan 25
Editorial Decision:	7th Feb 25
Revision Received:	23rd Apr 25
Editorial Decision:	9th May 25
Revision Received:	15th May 25
Accepted:	23rd May 25

Editor: *Esther Schnapp*

Transaction Report:

Dear Dr. Norris,

Thank you for the submission of your manuscript to EMBO reports. We have now received the comments from 2 referees, which are pasted below. Given that both reports are in fair agreement that this ms is interesting and can be published after minor revisions, I am making a decision now in the interest of time.

As you will see, the referees acknowledge the quality of your work and that the findings are interesting. They only have rather minor comments that should all be addressed.

I would thus like to invite you to revise your manuscript with the understanding that the referee concerns must be fully addressed and their suggestions taken on board. Please address all referee concerns in a complete point-by-point response. Acceptance of the manuscript will depend on a positive outcome of a second round of review. It is EMBO reports policy to allow a single round of major revision only and acceptance or rejection of the manuscript will therefore depend on the completeness of your responses included in the next, final version of the manuscript.

We realize that it is difficult to revise to a specific deadline. In the interest of protecting the conceptual advance provided by the work, we recommend a revision within 3 months (10th May 2025). Please discuss the revision progress ahead of this time with the editor if you require more time to complete the revisions.

- 1) A data availability section providing access to data deposited in public databases is missing. If you have not deposited any data, please add a sentence to the data availability section that explains that.
- 2) Your manuscript contains statistics and error bars based on $n=2$. Please use scatter blots in these cases. No statistics should be calculated if $n=2$.

5) a complete author checklist, which you can download from our author guidelines <<https://www.embopress.org/page/journal/14693178/authorguide>>. Please insert information in the checklist that is also reflected in the manuscript. The completed author checklist will also be part of the RPF.

6) Please note that all corresponding authors are required to supply an ORCID ID for their name upon submission of a revised manuscript (<<https://orcid.org/>>). Please find instructions on how to link your ORCID ID to your account in our manuscript tracking system in our Author guidelines <<https://www.embopress.org/page/journal/14693178/authorguide#authorshipguidelines>>

7) Before submitting your revision, primary datasets produced in this study need to be deposited in an appropriate public database (see <https://www.embopress.org/page/journal/14693178/authorguide#datadeposition>). Please remember to provide a reviewer password if the datasets are not yet public. The accession numbers and database should be listed in a formal "Data Availability" section placed after Materials & Method (see also <https://www.embopress.org/page/journal/14693178/authorguide#datadeposition>). Please note that the Data Availability Section is restricted to new primary data that are part of this study. * Note - All links should resolve to a page where the data can be accessed. *
If your study has not produced novel datasets, please mention this fact in the Data Availability Section.

12) All Materials and Methods need to be described in the main text using our 'Structured Methods' format, which is required for all research articles. According to this format, the Methods section includes a separate Reagents and Tools Table file (listing key reagents, experimental models, software and relevant equipment and including their sources and relevant identifiers) followed by a Methods and Protocols section describing the methods using a step-by-step protocol format. The aim is to facilitate adoption of the methodologies across labs. More information on how to adhere to this format as well as a downloadable template (.docx) for the Reagents and Tools Table can be found in our author guidelines: <https://www.embopress.org/page/journal/14693178/authorguide#structuredmethods>.

An example of a Method paper with Structured Methods can be found here: <https://www.embopress.org/doi/full/10.1038/s44320-024-00037-6#sec-4>

I look forward to seeing a revised form of your manuscript when it is ready.

Yours sincerely,

Referee #1:

Choudhary et al. have generated a fluorescent two-color splicing reporter for the *unc-13* microexon and uncovered neuron-type-specific microexon inclusion. In addition to PRP-40, a component of the U1 snRNP, they demonstrate that EXC-7 and MBL-1 cooperatively facilitate *unc-13* microexon inclusion by binding to adjacent sites in the downstream intronic region. Furthermore, the authors provide evidence of genetic and physiological interactions between PRP-40 and the RNA-binding proteins (RBPs). By employing genome editing to enforce microexon inclusion (*UNC-13included*) or skipping (*UNC-13skipped*), they further demonstrate that (i) *unc-13* microexon skipping is required for chemotaxis regulated by the AWA neuron, and (ii) *unc-13* microexon inclusion is essential for locomotory behavior and neuromuscular synaptic transmission, particularly in inhibitory motor neurons. Notably, the presence and regulation of this microexon appear to be conserved across the *unc-13* gene family in nematodes, as well as in one homolog in mice, suggesting a conserved role in nervous system function.

Overall, this study is well-structured, the data are clear, and the manuscript is well-written. The authors effectively demonstrate the biological significance of neuron-type-specific microexon inclusion and exclusion through genome editing *in vivo*. Particularly striking is their finding that the hypersensitivity of *exc-7; mbl-1* double mutants to aldicarb can be rescued by forced inclusion of a single microexon in a specific neuron class. This highlights the functional significance of this microexon as a key target of EXC-7 and MBL-1 among other alternative exons.

Specific Comments

1. Page 5, third paragraph, last two lines: The statement, "EXC-7 and MBL-1 cooperate in non-additive ways to facilitate *unc-13* microexon inclusion," as well as similar descriptions in the subsequent text, may not be appropriate for non-quantitative results. Even though the Δ PSI values (Figure 7G) suggest a measurable effect, they may not be linearly correlated with RBP function or the effects of RBP loss. I suggest rephrasing to more precisely describe the nature of the observed interaction.
2. Page 9, first paragraph, lines 3-4: The statement, "the *prp-40* mutant phenotype (complete microexon skipping) is dominant over either the *exc-7* or *mbl-1* mutant phenotypes," could be refined for clarity. Since the *prp-40* mutant already exhibits complete microexon skipping (GFP expression), no further reduction is possible. As a result, it is expected that the *exc-7; prp-40* and *prp-40; mbl-1* double mutants display the same phenotype as the *prp-40* single mutant. Typically, epistasis analysis is more informative when comparing mutants with opposing phenotypes.
3. Page 9, first section: The manuscript first describes genetic interactions, followed by physiological interactions. It may improve readability to reverse this order by presenting the physiological interaction of MBL-1 and PRP-40 *in vivo* (third paragraph) before discussing genetic interactions (first and second paragraphs). Since EXC-7 has already been shown to interact with PRP-40, this revised structure would provide a more logical flow of evidence supporting the proposed model.

Minor Points

1. Page 5, third paragraph, lines 3-4: In the sentence, "*mbl-1* mutation results in an decrease in microexon inclusion," the article "an" should be corrected to "a."
2. Page 8, Figure 3A: The orange line above the downstream intron appears to be separated into two segments. If this represents a bipartite deletion, please clarify this in the Figure legend.

Referee #2:

In this very impressive paper, Choudhary et al carefully study the regulation of microexon inclusion in an important synaptic protein, unc-13. Using elegant and rigorous analysis of not just RNAseq data, but also in vivo splicing reporters, the authors discover cell-type specific splicing patterns, identify regulators for these patterns and demonstrate the functional relevance of these splicing pattern.

Both breadth and depth of this analysis are impressive and I'm hesitant, but nevertheless insistent in pointing out one glaring issue that the authors should consider addressing - and that's the expression pattern of exc-7 and mbl-1. This is a relevant request because one big picture questions is whether exc-7 and mbl-1 are together sufficient to explain microexon splicing patterns and/or whether other factors remaining to be identified and/or whether things don't presently make sense. Notably, the authors of this paper had described expression patterns of exc-7 and mbl-1 in previous papers and there are good reporters out there (fosmid-based and reporter alleles). It appears that exc-7 and mbl-1 are co-expressed in cholinergic MNs but NOT in GABAergic neurons - and that's based simply on published data. Yet the authors are remarkably vague in describing the changes of the microexon splicing reporter in VNC MNs in exc-7/mbl-1 double mutants - are effects restricted to cholinergic neurons? I'm suspicious that this may not be the case because of the authors reluctance to raise this issue - but I do think this deserves further discussion. I find the discussion of the authors in the Discussion section not helpful at all -> they only refer to scRNA data ("This model is supported by single-cell expression data [50] indicating that olfactory neurons (e.g. AWA, AWB) exhibit low or undetectable levels of both RBPs, while motor neurons (e.g. DA, DB) exhibit high expression levels"). Why not referring to the reporter data? It would also be nice to explicitly confirm with the reporter the absence in sensory neurons (just pick the dye-fillable ones).

Two minor issues:

Page 2 & Fig.1 legend: Ref 24 -> change to Ref 22; make sure to reference updated bioRxiv and update further once in proofs (the paper is currently at eLife).

Page 5: please state if molecular nulls

Referee #1:

Choudhary et al. have generated a fluorescent two-color splicing reporter for the *unc-13* microexon and uncovered neuron-type-specific microexon inclusion. In addition to PRP-40, a component of the U1 snRNP, they demonstrate that EXC-7 and MBL-1 cooperatively facilitate *unc-13* microexon inclusion by binding to adjacent sites in the downstream intronic region. Furthermore, the authors provide evidence of genetic and physiological interactions between PRP-40 and the RNA-binding proteins (RBPs). By employing genome editing to enforce microexon inclusion (*UNC-13included*) or skipping (*UNC-13skipped*), they further demonstrate that (i) *unc-13* microexon skipping is required for chemotaxis regulated by the AWA neuron, and (ii) *unc-13* microexon inclusion is essential for locomotory behavior and neuromuscular synaptic transmission, particularly in inhibitory motor neurons. Notably, the presence and regulation of this microexon appear to be conserved across the *unc-13* gene family in nematodes, as well as in one homolog in mice, suggesting a conserved role in nervous system function.

Overall, this study is well-structured, the data are clear, and the manuscript is well-written. The authors effectively demonstrate the biological significance of neuron-type-specific microexon inclusion and exclusion through genome editing *in vivo*. Particularly striking is their finding that the hypersensitivity of *exc-7; mbl-1* double mutants to aldicarb can be rescued by forced inclusion of a single microexon in a specific neuron class. This highlights the functional significance of this microexon as a key target of EXC-7 and MBL-1 among other alternative exons.

Specific Comments

1. Page 5, third paragraph, last two lines: The statement, "EXC-7 and MBL-1 cooperate in non-additive ways to facilitate *unc-13* microexon inclusion," as well as similar descriptions in the subsequent text, may not be appropriate for non-quantitative results. Even though the Δ PSI values (Figure 7G) suggest a measurable effect, they may not be linearly correlated with RBP function or the effects of RBP loss. I suggest rephrasing to more precisely describe the nature of the observed interaction.

We have now adjusted the text (lines 143-146) to match the qualitative nature of the data, removing the quantitative-sounding terminology of "additive" and "synergistic." We have also tried to make the language more precise for Figure 7, where actual quantitative data is presented (lines 399-405).

2. Page 9, first paragraph, lines 3-4: The statement, "the *prp-40* mutant phenotype (complete microexon skipping) is dominant over either the *exc-7* or *mbl-1* mutant

phenotypes," could be refined for clarity. Since the prp-40 mutant already exhibits complete microexon skipping (GFP expression), no further reduction is possible. As a result, it is expected that the exc-7; prp-40 and prp-40; mbl-1 double mutants display the same phenotype as the prp-40 single mutant. Typically, epistasis analysis is more informative when comparing mutants with opposing phenotypes.

We agree that the 'epistasis' experiments mentioned (Fig 3F) are fairly unsurprising. We have now modified the language, removing the word epistasis entirely, and simply concluding that the results are consistent with the already-discussed notion that PRP-40 recruitment is the necessary downstream molecular event required for microexon inclusion (lines 205-211), as supported by the more-informative biochemical and trans-heterozygote experiments.

3. Page 9, first section: The manuscript first describes genetic interactions, followed by physiological interactions. It may improve readability to reverse this order by presenting the physiological interaction of MBL-1 and PRP-40 in vivo (third paragraph) before discussing genetic interactions (first and second paragraphs). Since EXC-7 has already been shown to interact with PRP-40, this revised structure would provide a more logical flow of evidence supporting the proposed model.

We have rearranged the ordering of this section according to the reviewer's suggestions (Lines 196-220, Figure 3F-H, and Figure 3 legend)

Minor Points

1. Page 5, third paragraph, lines 3-4: In the sentence, "mbl-1 mutation results in an decrease in microexon inclusion," the article "an" should be corrected to "a."

Fixed!

2. Page 8, Figure 3A: The orange line above the downstream intron appears to be separated into two segments. If this represents a bipartite deletion, please clarify this in the Figure legend.

Yes, this is indeed a bipartite deletion, and we now clarify this in the figure legend (Lines 184-187).

Referee #2:

In this very impressive paper, Choudhary et al carefully study the regulation of microexon inclusion in an important synaptic protein, *unc-13*. Using elegant and rigorous analysis of not just RNAseq data, but also in vivo splicing reporters, the authors discover cell-type specific splicing patterns, identify regulators for these patterns and demonstrate the functional relevance of these splicing pattern.

Both breadth and depth of this analysis are impressive and I'm hesitant, but nevertheless insistent in pointing out one glaring issue that the authors should consider addressing - and that's the expression pattern of *exc-7* and *mbl-1*. This is a relevant request because one big picture questions is whether *exc-7* and *mbl-1* are together sufficient to explain microexon splicing patterns and/or whether other factors remaining to be identified and/or whether things don't presently make sense. Notably, the authors of this paper had described expression patterns of *exc-7* and *mbl-1* in previous papers and there are good reporters out there (fosmid-based and reporter alleles). It appears that *exc-7* and *mbl-1* are co-expressed in cholinergic MNs but NOT in GABAergic neurons - and that's based simply on published data. Yet the authors are remarkably vague in describing the changes of the microexon splicing reporter in VNC MNs in *exc-7/mbl-1* double mutants - are effects restricted to cholinergic neurons? I'm suspicious that this may not be the case because of the authors reluctance to raise this issue - but I do think this deserves further discussion. I find the discussion of the authors in the Discussion section not helpful at all -> they only refer to scRNA data ("This model is supported by single-cell expression data [50] indicating that olfactory neurons (e.g. AWA, AWB) exhibit low or undetectable levels of both RBPs, while motor neurons (e.g. DA, DB) exhibit high expression levels"). Why not referring to the reporter data? It would also be nice to explicitly confirm with the reporter the absence in sensory neurons (just pick the dye-fillable ones).

Thank you to the reviewer for pointing out this important matter that we did not do justice to in the manuscript. We have now analyzed both transcriptome data and fosmid reporters, and have added the following observations to the manuscript:

1. Indeed, ALL ventral cord motor neurons (both cholinergic and GABAergic) express only the skipped isoform in *exc-7; mbl-1* double mutants (added in line 142-143).
2. Revisiting our fosmid reporters generated for Thompson *et al.*, 2019, we find that indeed, *exc-7* expression is highly restricted to cholinergic, but not GABAergic, ventral cord motor neurons (now added to fig S6, and lines 462-465). *mbl-1*, however, while its signal is very bright in cholinergic neurons, is also expressed at low

but detectable levels in GABAergic neurons as well (added to fig S6). This is in contrast to *exc-7*, where no signal at all is detected in GABA motor neurons. Transcriptomic data from CeNGEN and other single-cell sequencing experiments also support this finding, with *exc-7* levels being high in cholinergic and absent in GABAergic, while *mb1-1* levels are high in both cholinergic and GABAergic motor neurons (Fig S6, and lines 462-465). Finally, we note in passing that the *mb1-1* fosmid used is a C-terminal protein-RFP fusion, and there is evidence for complex alternative C-terminal *mb1-1* isoforms (both exon skipping and intron retention) that could further contribute to the complex relationship between the fosmid expression levels, the transcriptomic data, and the functional cell-specific splicing regulation controlled by MBL-1.

3. Focusing on olfactory neurons, we provide evidence both from the fosmid reporters and the transcriptomic data that olfactory neurons in the head exhibit no/low levels of *exc-7* and *mb1-1* expression, while motor neurons exhibit high levels of both (Fig S6, and lines 462-465).

Two minor issues:

Page 2 & Fig.1 legend: Ref 24 -> change to Ref 22; make sure to reference updated bioRxiv and update further once in proofs (the paper is currently at eLife).

We have now changed the references as requested, and since the preprint is now reviewed and publicly available at eLife, we have updated to the current eLife version.

Page 5: please state if molecular nulls

This information has now been added (Lines 154-155).

Dear Dr. Norris,

Thank you for the submission of your revised manuscript. We have now received the enclosed comment from referee 2, who was asked to assess it, and I am happy to say that we can in principle accept your ms.

Only a few editorial requests will need to be addressed:

- Please remove the figures from the ms file. The figure legends should be kept at the end of the ms - first the main figure legends should be listed and then EV figure legends.
- The Disclosure and competing interest statement is provided twice: the one before the Acknowledgments is the correct one, but needs to be placed after the Acknowledgments.
- The REFERENCE format needs to be correct. It needs to be alphabetical, not numerical; et al needs to be used after 10 author names. Please use the EMBO reports style.
- The author CHECKLIST is missing responses to the pull down menus in cells D106-D108, please complete and send us the completed checklist.
- The individual EV Figures have incorrect titles - Figure S1, etc. They need to be corrected to Figure EV1, etc.
- The Reagents & Tools TABLE needs to be removed from the ms and uploaded as a separate file.
- Our routine figure check of revised ms detected some potential image duplications that are not explained in the figure legends:
Cell reuse between Figure 1F and Figure 2D.
Cell reuse between Figure 1F and Appendix Fig S4A.
Cell reuse between Figure 2D and Appendix Fig S4A.
Can you please clarify what happened? If these are indeed the same images, this needs to be justified and explained in the figure legends.

Figure Legends - Comments

- Please note that the exact p values are not provided in the legends of figures 4C, F, H; 5B, C, G, H; 6B, D, F, G, I, J; 7H. Please provide exact p-values are reasonable.
- Please indicate the statistical test used for data analysis in the legends of figures 4C, E, F, H; 5B, C, G, H; 6B, D, F, G, I, J; 7H
- Please note that the box plots need to be defined in terms of minima, maxima, centre, bounds of box and whiskers, and percentile in the legend of figure 7D
- Please note that information related to n is missing in the legends of figures 1C, 2A
- Please note that the error bars are not defined in the legends of figures 1C, 2A
- Please note that the white arrow heads are not defined in the legend of figure 4G, EV1 C. This needs to be rectified.

- Please mention in the title or abstract that this study is done on *C. elegans*.

EMBO press papers are accompanied online by A) a short (1-2 sentences) summary of the findings and their significance, B) 2-3 bullet points highlighting key results and C) a synopsis image that is exactly 550 pixels wide and 200-600 pixels high (the height is variable). The synopsis image should provide a sketch of the major findings, like a graphical abstract. Please note that text needs to be readable at the final size. Please send us this information along with the final manuscript.

Referee #2:

The authors have provided satisfying responses to my comments and I find the paper ready for publication, as is.

Greetings Esther,

Thank you for the comments. We believe we have now addressed all of the items you mentioned. New files are attached, and new text in the manuscript file is in red (text in response to the reviewers' previous comments remains in blue).

Sincerely,

Adam Norris

Adam D Norris
University of California, Riverside
United States

Dear Dr. Norris,

I am very pleased to accept your manuscript for publication in the next available issue of EMBO reports. Thank you for your contribution to our journal.
